# Ecosystem-based fisheries management forestalls climate-driven collapse

K. K. Holsman [1,2 ✉], A. C. Haynie[1], A. B. Hollowed[1,2], J. C. P. Reum[1,2,3], K. Aydin[1,2], A. J. Hermann [4,5], W. Cheng [4,5], A. Faig [2], J. N. Ianelli[1,2], K. A. Kearney [1,4] & A. E. Punt [2]

Climate change is impacting fisheries worldwide with uncertain outcomes for food and nutritional security. Using management strategy evaluations for key US fisheries in the eastern Bering Sea we find that Ecosystem Based Fisheries Management (EBFM) measures forestall future declines under climate change over non-EBFM approaches. Yet, benefits are species-specific and decrease markedly after 2050. Under high-baseline carbon emission scenarios (RCP 8.5), end-of-century (2075–2100) pollock and Pacific cod fisheries collapse in >70% and >35% of all simulations, respectively. Our analysis suggests that 2.1–2.3 °C (modeled summer bottom temperature) is a tipping point of rapid decline in gadid biomass and catch. Multiyear stanzas above 2.1 °C become commonplace in projections from ~2030 onward, with higher agreement under RCP 8.5 than simulations with moderate carbon mitigation (i.e., RCP 4.5). We find that EBFM ameliorates climate change impacts on fisheries in the near-term, but long-term EBFM benefits are limited by the magnitude of anticipated change.

[1] National Oceanic and Atmospheric Administration, Alaska Fisheries Science Center, 7600 Sand Point Way N.E., Seattle, WA 98115, USA. [2] School of Aquatic and Fishery Sciences, University of Washington, Seattle, WA 98195, USA. [3] Institute for Marine and Antarctic Studies and Centre for Marine Socioecology, University of Tasmania, Hobart TAS 7001, Australia. [4] Joint Institute for the Study of the Atmosphere and Ocean, now Cooperative Institute for Climate, Ocean, and Ecosystem Studies, University of Washington, Seattle, WA 98195, USA. [5] Ocean Environment Research Division, NOAA/Pacific Marine Environmental Laboratory, Seattle, WA 98115, USA. ✉email: kirstin.holsman@noaa.gov

Marine ecosystems face an unknown future[1–5]. Multiple studies predict large ecosystem reorganization concomitant with future climate change[6]; postulations which are increasingly observed in a variety of marine ecosystems[1,2,4]. While evidence of potential and realized climate change impacts in marine systems is widespread, implementation of climate-adaptive strategies for maritime societies and economies is less commonplace[7,8]. This reflects the naturally dynamic nature of marine systems and the challenge of designing and implementing policies that can address impacts and risk from both rapid and chronic climate-driven change. Marine capture fisheries are especially vulnerable to climate change impacts[5,9–11] as marine organisms are often sensitive to small shifts in ocean temperature, circulation, and chemistry[3,12].

Presently, climate-adaptive measures are largely missing from fisheries management policies and approaches[5,13,14]. Intergovernmental and national climate assessments have highlighted the need to evaluate existing fishery management plans for maladaptation to climate change[4,5,15]. Most of these assessments point to an ecosystem management (EM) approach to promote resilient marine ecosystems and fisheries[13,16,17]. EM ranges from an ecosystem approach to single-species management (i.e., EM as context for management focused on optimizing a single species) to full ecosystem-based management (i.e., EM applied across sectors to manage the entire ecosystem). Ecosystem-based fishery management (EBFM, i.e., EM applied to the fishery sector) is intermediate to these approaches, and expands classic adaptive management strategies to additionally utilize ecosystem information to manage multiple species across the ecosystem[16,18]. Intuitively, the more holistic EBFM approach should impart climate resilience to fisheries, yet few studies have demonstrated the performance of EBFM under climate change (but see ref. [17]).

Here we use scenario analyses and management strategy evaluation (Fig. 1)[19,20] to assess the future performance of EBFM fisheries policies as implemented in the Eastern Bering Sea, Alaska, for the past two decades[21]. This highly productive system supports the largest fishery in the United States (walleye pollock,

*Gadus chalcogrammus*) with ~1.4 million ton yr$^{-1}$ and \$1.34 billion USD first wholesale value in 2017. Pacific cod (*G. microcephalus*) is also important in this region and is one of the most economically valuable groundfish fisheries in the USA[22]. These fisheries operate under policies that are among the most well established and successful examples of fisheries EBFM[21]. A key feature of regional EBFM is an over-arching 2 million ton annual combined groundfish catch limit (hereafter 2 MT cap) aimed at preserving ecosystem function[21]. Managers reduce annual harvest limits for individual stocks to conform to the 2 MT cap based on multiple management objectives, including maximizing sustainable yield, reducing the risk of exceeding directed and incidental catch limits (which can close a fishery for the season), other ecosystem considerations and impacts, and meeting distributional objectives and mandates[21].

Current EBFM in the eastern Bering Sea has sustained high fisheries yield over the last three decades despite considerable environmental variability and with few instances of overfishing[21,23]. However, the management system is yet untested against the unidirectional and potentially large changes anticipated under climate change. Indeed, recent and extreme warming and loss of sea ice in the Eastern Bering Sea (especially during recent unprecedented multiyear marine heatwaves between 2014 and 2019) has led to the rapid poleward redistribution of Pacific cod and declines in recruitment and productivity of several groundfish species[24–26]. Observed recent warming, sea ice loss, and biophysical responses in the Bering Sea[27,28] are consistent with previous projections of impacts of climate change, yet were not anticipated to manifest until mid-century[29]. Marine species have exhibited responses that are both consistent with predictions (e.g., rapid northward distributional shifts of multiple benthic species, declines in fish recruitment, declines in large lipid-rich zooplankton species)[29,30] and unanticipated (e.g., near-term climate resilience of pollock[26] or sudden widespread sea bird mortality events[31]).

The most recent biophysical projections[29] indicate that further warming and reduced lower trophic level production in the

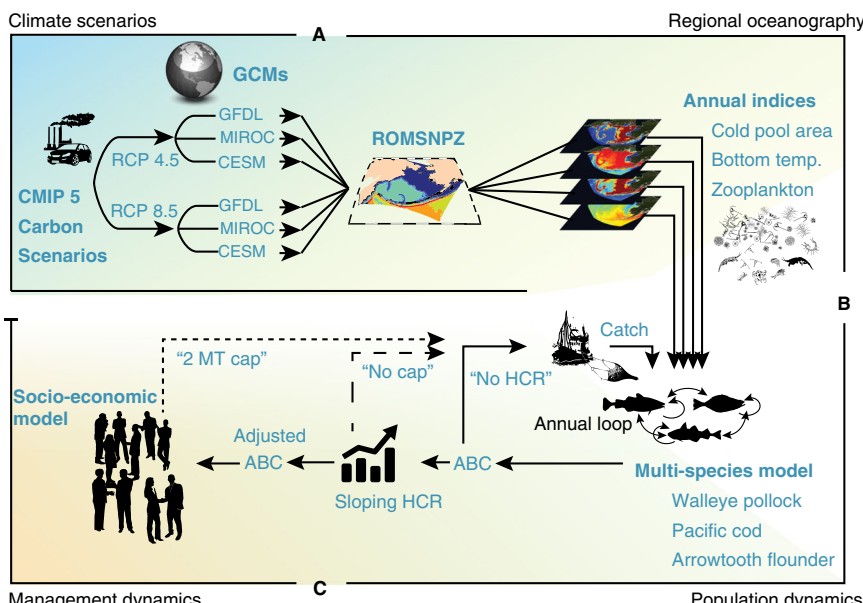

**Fig. 1 Model coupling framework. a** Regional downscaling where three global climate models driven by the IPCC AR5 CMIP5 emission scenarios determine boundary conditions of the coupled ROMSNPZ high resolution oceanographic model for the Bering Sea, AK. **b** Biological downscaling of annual indices from the ROMSNPZ were used to drive thermal parameters in the CEATTLE model (i.e., weight-at-age and predation) as well as climate-enhanced spawner-recruitment relationships. **c** Annual harvest recommendations (ABC) from the assessment model which were translated into annual catch using the ATTACH social-economic model of the effect of EBFM policies on harvest.

Bering sea is probable, with uncertain outcomes for major fisheries. Previous simulation studies in the region have focused on individual stocks and ignored inter-species and inter-fishery interactions or evaluated multispecies or multi-fleet projections without also resolving climate impacts on population dynamics. Consequently, they provide limited utility in evaluating the performance of status quo EBFM policies under climate change (but see refs. [19,32]). Here we use coupled climate-enhanced multi-species assessment and fishery management models to evaluate if EBFM reduces future risk of fishery declines, alters thermal tipping points for fisheries, and imparts increased stability and sustainability in harvest under climate change.

We find that EBFM policies help ameliorate climate change impacts on fisheries in the eastern Bering Sea in the near-term, yet benefits are limited after mid-century when climate-driven declines exceed adaptive capacity. Under representative concentration pathway (RCP) 8.5 (i.e., high-baseline carbon emissions[33]), end-of-century (2075–2100) pollock and Pacific cod fisheries collapse in >70 and >35% of all simulations, respectively. Our analysis suggests that 2.1–2.3 °C (modeled summer bottom temperature) is a tipping point of rapid decline in gadid biomass and catch across climate emission scenarios and management approaches investigated. Multiyear stanzas above 2.1 °C become commonplace in projections from ~2030 onward, with higher agreement under RCP 8.5 than simulations with moderate carbon mitigation (i.e., RCP 4.5). We therefore conclude that EBM represents a climate-resilient approach for managing marine living resources in the near-term, but with the caveat that adaptation through EBFM is limited by socioeconomic constraints and the magnitude of change anticipated by mid-century under high emission scenarios.

## Results

**Projected changes in environmental conditions.** Increases in summer bottom temperature indices are projected for the EBS under both representative concentration pathway (RCP) 4.5 and 8.5, but are consistently highest for RCP 8.5 (i.e., high-baseline carbon emissions). Under RCP 4.5 (i.e., moderate carbon mitigation), two of the three GCMs project warming of ~1–2.5 °C over the next 80 years, and increases of 2–4.5 °C are projected under RCP 8.5 for all three GCMs by end of the century (Fig. 2).

**Projected changes in unfished spawning biomass.** Under all 3 RCP 8.5 projections, and 2 of 3 RCP 4.5 runs, the combined

effects of increased metabolic demand, reduced availability of lipid-rich prey[26,29] (Supplementary Fig. 1), and increased overlap with juvenile gadid predators, resulted in reduced survival and overwintering success of juvenile gadids and led to long-term declines in groundfish populations (Figs. 3, S2). Unfished spawning biomass for pollock and cod declined under both RCP 4.5 and 8.5 projection scenarios, with greater and more consistent declines projected for pollock and cod under RCP 8.5 (Fig. 3b, d). Relative to the persistence scenario (where future climate was held constant at average 2006–2017 hindcast conditions), under RCP 4.5 and RCP 8.5, end-of-century (2075–2100) unfished pollock spawning stock biomass declined on average by 47% and 70%, respectively, cod declined 23% and 41%, respectively, and arrowtooth flounder increased 7% and declined 6%, respectively. Notably, under RCP 8.5 more than a third of all simulations resulted in >90% declines in pollock unfished spawning biomass by end-of-century (relative to the persistence scenario).

**Effect of the EBFM 2 MT cap on projected catch.** In the absence of the EBFM 2 MT cap, declines in fished spawning stock biomass were projected for all species (Supplementary Fig. 2). Implementation of the 2 MT cap stabilized pollock catch (up to mid-century) and ameliorated climate-driven declines in pollock catch and biomass in all scenarios, despite substantial variation among projections (Figs. 4, S2). These effects were greatest for scenarios where warming was minimal and spawning biomass remained relatively high; benefits of the 2 MT cap were reduced after mid-century in projections with significant warming.

The 2 MT cap had little effect on Pacific cod, which is managed close to the target fishing mortality rate (Supplementary Fig. 3). However, to provide for other groundfish fisheries under the 2 MT cap, fishing mortality for pollock at high stock sizes was reduced when stocks were above the target spawning biomass[34] (https://github.com/amandafaig/catchfunction) resulting in a reduction in the spawning exploitation rate (Supplementary Fig. 3). A similar pattern of reduced exploitation rate at higher biomass was also emergent for arrowtooth flounder simulations (Supplementary Fig. 3) resulting in a large benefit of the 2 MT cap for arrowtooth flounder catch, especially in RCP 8.5 scenarios where catch under the 2 MT cap was stable as compared to declines in scenarios without the 2 MT cap (Fig. 4).

The effect of the EBFM 2 MT cap on pollock catch and biomass varied with the magnitude of climate-driven change. The 2 MT cap reduced the risk of decline in catch or biomass relative to the same scenarios without the 2 MT cap early in the

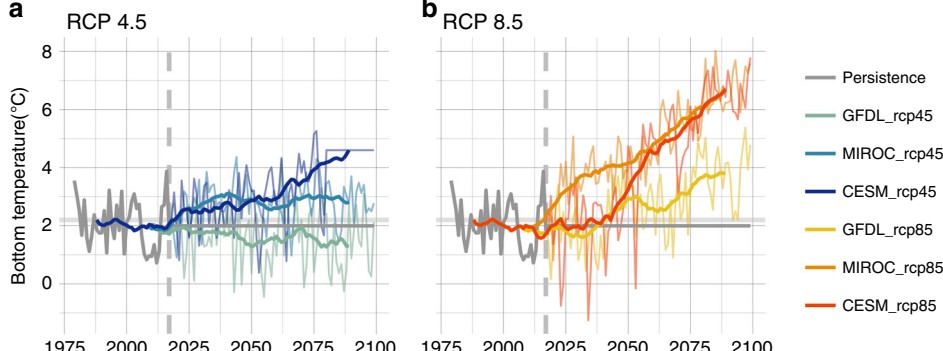

**Fig. 2 Future Bering Sea bottom temperatures relative to the ~2.1 OC tipping point.** Bias-corrected projections of survey replicated annual mean summer bottom temperature (°C) for the Bering Sea under CMIP5 Representative Concentration Pathway (RCP) 4.5 (**a**) and RCP 8.5 (**b**). Annual temperatures (thin lines) and 20 year running mean (thick lines) temperatures are based on survey replicated samples from the downscaled ROMSNPZ Bering10K model. Persistence is based on 2006–2017 average conditions from the downscaled ROMSNPZ hindcast (gray line). Vertical dashed lines represent the start of the projection period (2018). Projections include three global climate models; the Geofluid Dynamics Lab Earth System Model (GFDL-ESM2M), the MIROC-ESM, and the National Center for Atmospheric Research Community Earth System Model (CESM1). See ref. [29] for more detail.

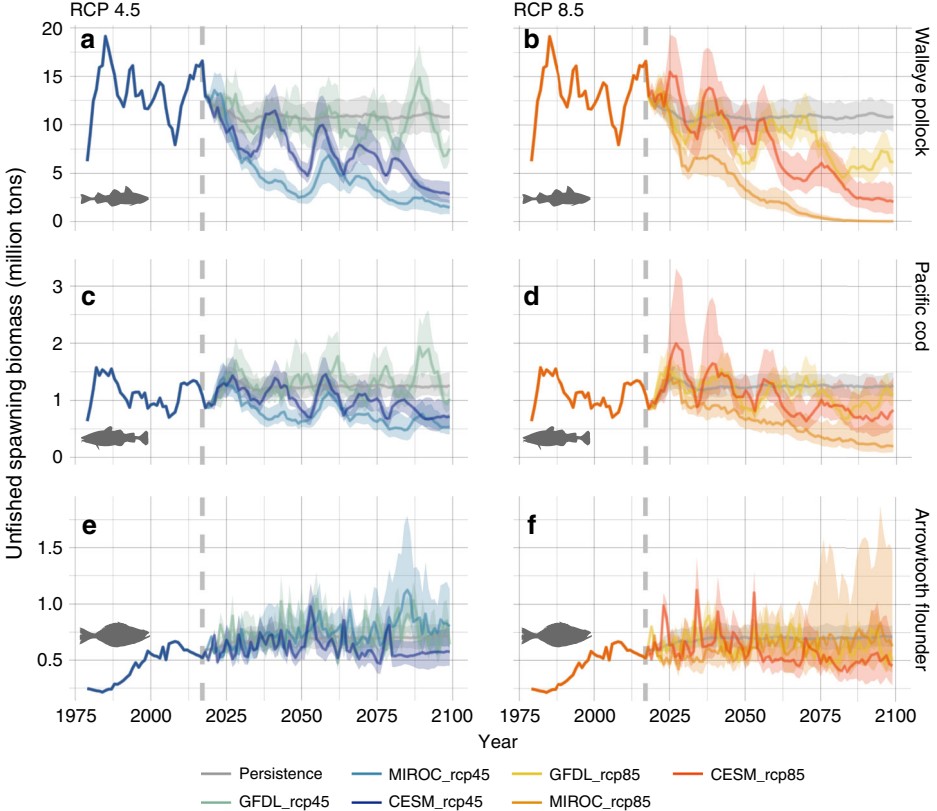

**Fig. 3 Future unfished spawning stock biomass.** Unfished spawning stock biomass for pollock (**a**, **b**), Pacific cod (**c**, **d**), and arrowtooth flounder (**e**, **f**) under future climate change scenarios: moderate mitigation (RCP 4.5; left column), high-baseline emissions (RCP 8.5; right column), and a persistence baseline climatology (solid gray lines in each panel). Vertical dashed line represents the start of the projection period (2018–2100). Solid lines represent the 50th quantile, while shading indicates the 10th and 90th quantiles from 100 random draws from estimated recruitment parameters.

projection period for RCP 8.5 for pollock (Fig. 5a, b; Supplementary Fig. 4A). However, risk of decline increased within each successive 25-yr period and there was no reduction of risk after 2080 under the 2 MT cap when declines were imminent regardless of EBFM policies. The beneficial effect of the 2 MT cap on reducing risk of declines in catch was greater under moderate mitigation scenarios (RCP 4.5) relative to the high baseline (RCP 8.5) for pollock. Yet, regardless of management scenario (i.e., with or without the 2 MT cap), under RCP 8.5 roughly 70% of simulations resulted collapse of pollock catch (i.e., >80% decline relative to the persistence scenario) by end-of-century and ~90% of simulations exhibited severe declines in catch (i.e., >50% decline).

**Risk of fishery collapse under climate change.** Similar to pollock, the 2 MT cap also reduced the risk of decline (i.e., >10% decline relative to the persistence scenario) in arrowtooth flounder catch, yet unlike pollock, reduction in risk of declines in arrowtooth catch under the 2 MT cap was greater under RCP 8.5 than RCP 4.5 (Supplementary Table 1). The 2 MT cap had little effect on Pacific cod catch in general, thus reduction in risk of declines in Pacific cod catch under the 2 MT cap (as compared to no cap scenarios) were marginal (Supplementary Fig. 4). Across all scenarios, by end-of-century roughly a third of simulations exhibited collapse of Pacific cod catch and ~65% of simulations resulted in severe declines in catch (i.e., >50%).

**Thermal tipping points for fishery collapse.** Threshold analysis suggests that a summer survey average bottom temperature of 2.1–2.3 °C is a tipping point for changes in catch (relative to the

persistence scenario) from stable (or increasing) to rapid decline for Pacific cod and pollock (Fig. 6, Supplementary Fig. 5). In contrast to scenarios without the 2 MT cap, warming is associated with an increase (rather than a decrease) in arrowtooth catch relative to the climate persistence scenario. Multiyear warm stanzas with five consecutive years above the putative 2.1 °C threshold occurred in only one period of the hindcast (2004–2005) but become commonplace in projections from 2033 onward in all three models under RCP 8.5, and two of the models under RCP 4.5 (Fig. 2, Supplementary Fig. 6).

**Discussion**
EBFM policies, such as the system-wide 2 MT cap on cumulative annual groundfish harvest (and the attendant reduction in exploitation rate at high stock sizes) may forestall declines in biomass and catch under climate change and provide fisheries and fishers a critical window of opportunity to prepare for and adapt to change. Implemented over 35 years ago, the 2 MT cap was not specifically designed to enable resilience to climate change. However, it provides an important stabilizing role in maintaining consistently high catches and spawning biomass of pollock in the EBS despite climate-driven change. That said, we also found that climate-driven changes may exceed the adaptive capacity of current fisheries management after 2050 and significant declines of >80% (relative to the persistence scenario) were projected for pollock and cod after 2075 in most simulations under the warmest scenarios regardless of management approaches trialed.

EBFM policies interact with climate sensitivity to impact species differently. In this example, the EBFM 2 MT cap tips the balance for arrowtooth flounder such that future catch is stable

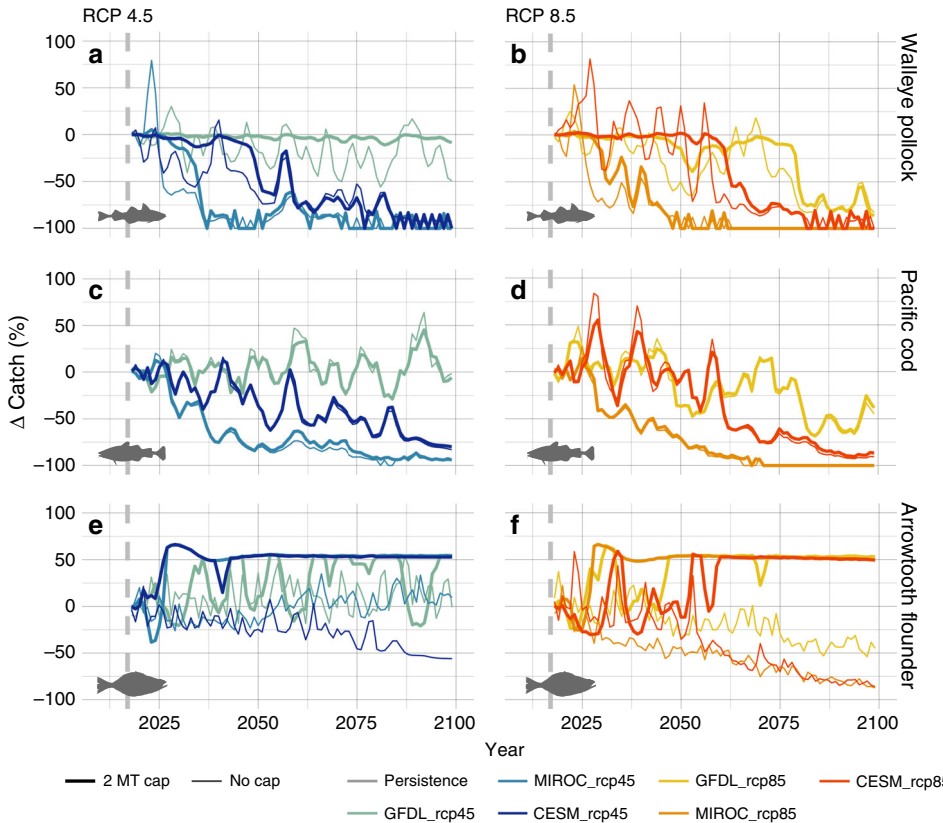

**Fig. 4 Change in catch (%) relative to the persistence scenario (constant climate).** Changes in catch relative to the persistence scenario for pollock (**a**, **b**), Pacific cod (**c**, **d**), and arrowtooth flounder (**e**, **f**) under future climate change scenarios: moderate mitigation (RCP 4.5; left column), high-baseline emissions (RCP 8.5; right column), and a persistence baseline climatology (solid gray lines in each panel). Lines represent management scenarios when catch is equal to the annual harvest limit ($ABC_y$) based on a sloping harvest control rule (i.e., no cap; thin lines) and when catch is equal to allocation using the same $ABC_y$ in combination with a 2 MT ecosystem cap on groundfish harvest (i.e., 2 MT cap; thick lines). Vertical dashed lines represent the start of the projection period (2018–2100).

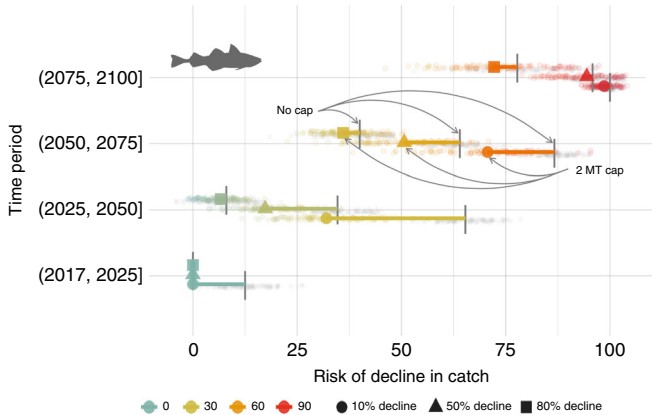

**Fig. 5 Change in risk of decline in pollock catch under climate change (RCP 8.5) relative to the climate persistence scenario.** Risk includes risk of decline (>10% decline), severe decline (>50% decline) and collapse (>80% decline) in catch during four time periods for scenarios without the cap (gray points) and those with the 2 MT cap. The color scale represents relative risk (0–100) from low (teal) to high (red). Length of segments indicate the magnitude of change in risk between the no cap and 2 MT cap scenarios. Vertical gray segments indicate the mean of the "no cap" scenarios.

and sometimes increases under future climate change. Yet for Pacific cod, the 2 MT cap EBFM policy has little effect on catch and therefore does not alter the outcome of declines under climate change. This reflects historical precedent of the effect of the

2 MT cap for maximizing catch of the most valuable species in the aggregate complex[34] (i.e., Pacific cod; https://github.com/amandafaig/catchfunction). Contrasting effects of the 2 MT cap across species suggest that EBFM harvest policies that reduce the spawning exploitation rate at high abundances should buffer stocks and reduce vulnerability to climate-induced collapse, at least in the near-term. Of note, the benefit of the EBFM 2 MT cap scales with rate of warming, with collapse occurring before 2050 in the fastest warming simulations even under the 2 MT cap.

In contrast to previous analyses[23,35] we do not find support for potential increased yield under climate change for Alaska fisheries. Our results are more consistent with regional and species-specific analyses that project declines for many Bering sea species[36,37]. The risk of declines in spawning biomass and catch for the species in our model increases between RCP 4.5 and 8.5 and over time, with projected declines diverging from the persistence scenario by 2050, even with the 2 MT cap. Prior to 2050, the general trend is a decline relative to the persistence scenario, but there is considerable variability among GCMs due to process error.

Our study identified a critical ~2.1–2.3 °C summer bottom temperature (i.e., ~0.2 °C warming relative to the 1979–2017 period) threshold for declines in catch that can inform regional fisheries management, especially if combined with climate forecasting tools[38]. Projected declines in fall zooplankton biomass are associated with increasing bottom temperatures (Supplementary Fig. 1)[29,30] and retrospective analyses also suggest that recent historical warm stanzas were associated with multi-trophic responses in the

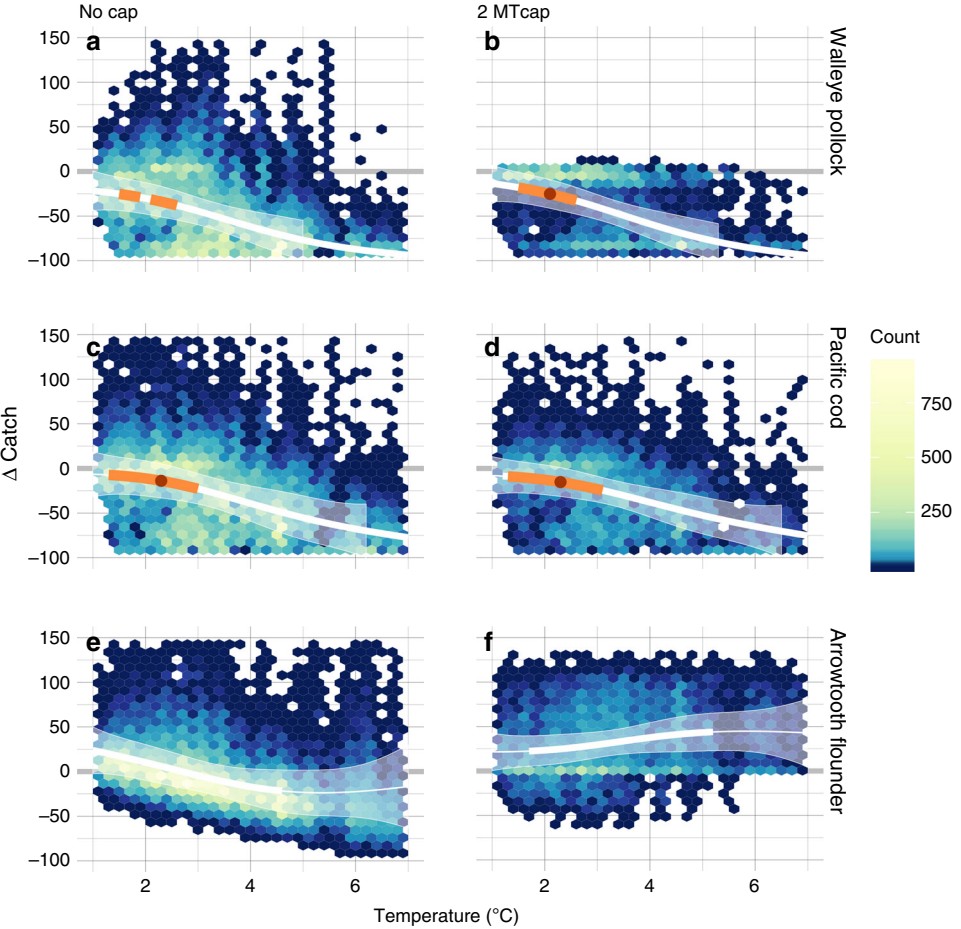

**Fig. 6 Thermal tipping points for catch and biomass.** Analysis of proportional change in future catch relative to the persistence scenario (ΔCatch) as a function of future bottom temperature (°C). Solid lines represent the mean smoothing function (s(x)); shading indicates the 2.5 and 97.5% quantiles from 1000 bootstrap replicates. Scenarios without the 2 MT cap (**a**, **c**, **e**); scenarios with the 2 MT cap (**b**, **d**, **f**). Rows correspond to each species. The thick white and orange lines indicate areas where the 95% CI of the first derivative (s(x)′) of the smoothing functions do not include zero; orange bar indicates indicate where the 95% CI of the second derivative (s(x)″) does not overlap zero; on each line, red circles indicate the best estimate of the tipping point (i.e., s(x)″ is most different from zero).

Bering Sea, including declines in fall zooplankton and reduced survival of upper-trophic level consumers[26,31,39]. Across projections, such conditions become commonplace in 2033, but occur as early as 2025 in some model trajectories (MIROC RCP 8.5) or not at all (GFDL RCP 4.5).

Few studies to date have confronted the assumption that tools developed to promote stability under the assumption of stationarity in biological processes will perform well under climate change. Current management tools may perform poorly in increasingly volatile climate conditions[40]. For example, catch shares programs that incentivize long-term stewardship through stock ownership increase the safe operating space for fisheries management, but can abruptly fail at low biomass levels[40]. Others have found potential for climate change to erode confidence in the management process even in well-managed systems[41]. Finally, management measures aimed at stabilizing resources over time can increase long-term instability[42–44]. Indeed, we found that the EMFB 2 MT cap stabilized fisheries in the near-term but increased the risk of sudden collapse of pollock catch at the end of the century (i.e., collapse occurs rapidly without preceding declines in catch). Concurrent climate-driven declines in spawning stock biomass preceded collapse, reinforcing the importance of fishery-independent estimates of biomass for early warnings of impending fishery declines. Hyperstability is an

inherent risk in decoupled harvest and biomass dynamics[45] and is a potential outcome of the EBFM cap which should be explored in future management strategy evaluations. Climate-resilient fisheries management may require transformative approaches that embrace variability rather than target biological stability[40,43,44] in order to ensure future fisheries and food security under climate change[46].

Increasing flexibility in fisheries management is often advocated as an approach to help fisheries adapt to climate-driven change[14,47,48]. We found that fixed management measures, especially those with an EBFM focus such as the 2 MT cap on harvest, also impart benefits to future fisheries through stabilizing catch, at least to a point. It appears new methods, or potentially adaptive methods, may be required as systems are put under increasing stress beyond mid-century. Sustainable harvest limits combined with sloping harvest control rules, a common single-species management tool, induced large oscillations in biomass that could destabilize fisheries when used alone (i.e., without an ecosystem 2 MT cap on total yield; Fig. 4). In practice, a portfolio of management approaches that integrate both flexible and fixed management measures is likely needed to promote adaptation across multiple scales of impact[13,14,17].

We focused our analyses on status quo EBFM policy performance under future conditions, yet these polices were not

implemented to specifically address climate change. Our framework enables consideration of a much broader set of scenarios, for instance, implementation of adaptive or climate-informed alternatives to the 2 MT cap, EBM policies that optimize ecosystem productivity (sensu[17]), or policies that favor climate-informed single-species management over current climate-naive ecosystem caps on total allowable catch (TAC). When considered with the current scenario suite, such scenarios could further characterize the effectiveness of EBFM in facilitating climate adaptation and represent important next steps for evaluation.

## Methods

**Regionally-downscaled climate change projections**. We used a management strategy evaluation (MSE) applied to ensemble projections of a climate-enhanced multispecies stock assessment within the integrated modeling framework of the Alaska Climate Change Integrated Modeling project (ACLIM)[19]. For this, six high resolution downscaled projections of oceanographic and lower trophic level conditions in the Bering Sea (using the Regional Ocean Modeling System[49,50]) were coupled to the BESTNPZ nutrient-phytoplankton-zooplankton model[51]; we refer to this model complex throughout this paper as the Bering10K ROMSNPZ, or just ROMSNPZ, model. Boundary conditions were driven by three global general circulation models (GFDL-ESM2M[52], CESM1[53], and MIROC-ESM[54]) projected (2006–2099) under the high-baseline emission scenario Representative Concentration Pathway 8.5 (RCP 8.5) and midrange global carbon mitigation (RCP 4.5; note, that for CESM under RCP 4.5, projections from 2080–2100 were unavailable so conditions from 2080–2099 were held constant at 2080 conditions for that scenario only) future scenarios from the Coupled Model Intercomparison Project phase 5 (CMIP5)[29,55]. Hermann et al.[30,56] also report on downscaled hindcasts of oceanographic and lower trophic conditions in the EBS from 1970–2012 (see refs. [30,56,57] for detailed descriptions of model evaluation and performance). For each downscaled model simulation, we replicated the National Marine Fisheries Service Alaska Fisheries Science Center annual summer bottom-trawl survey in time and space in the ROMSNPZ model (using historical mean survey date at each latitude and longitude of each gridded survey station) to derive estimates of sea surface and bottom temperatures (Fig. 1). We additionally used a polygon mask of the survey area to estimate the average zooplankton abundance in the system during spring, summer, winter, and fall months. These indices were derived for each climate projection scenario, as well as a persistence scenario where conditions were held constant at the average of those for 2006–2017 from a hindcast simulation. All index projections were bias corrected to the 2006–2017 hindcast period using the delta method assuming unequal variance in the GCM projections and hindcast[58] such that:

$$T'_{\text{fut},y} = \bar{T}_{\text{hind},\overrightarrow{\text{ref}}} + \frac{\sigma_{\text{hind},\overrightarrow{\text{ref}}}}{\sigma_{\text{fut},\overrightarrow{\text{ref}}}}\left(T_{\text{fut},y} - \bar{T}_{\text{fut},\overrightarrow{\text{ref}}}\right) \quad (1)$$

where $T'_{\text{fut},y}$ is the bias-corrected projected timeseries, $T_{\text{fut},y}$ is the raw projected timeseries, $\bar{T}_{\text{hind},\overrightarrow{\text{ref}}}$ is the mean of the hindcast during the reference years $\overrightarrow{\text{ref}}$ (2006–2017), $\bar{T}_{\text{fut},\overrightarrow{\text{ref}}}$ is the mean of the raw projected timeseries during the reference years $\overrightarrow{\text{ref}}$, $\sigma_{\text{hind},\overrightarrow{\text{ref}}}$ is the standard deviation of the hindcast during the reference years $\overrightarrow{\text{ref}}$, $\sigma_{\text{fut},\overrightarrow{\text{ref}}}$ is the standard deviation of the raw projection timeseries during the reference years $\overrightarrow{\text{ref}}$.

**Climate-enhanced multispecies stock assessment model**. Bias-corrected indices were then used as covariates in the climate-enhanced multispecies stock assessment model for the Bering Sea (hereafter CEATTLE)[59] to evaluate the performance of alternative management approaches on future fish biomass and catch. CEATTLE is a climate-enhanced multispecies statistical age-structured assessment model with parameters for growth that are functions of temperature (i.e., temperature-specific average weight-at-age) and predation that are functions of temperature (via a bioenergetics-based predation sub-model)[59–61]. Since 2016, the model has been used operationally in the Bering sea as a supplement to the annual BSAI pollock stock assessment[61]. Various configurations of CEATTLE are possible; for this study we chose one where temperature-specific predator and prey interactions influenced natural mortality, temperature influenced weight-at-age, and the spawner-recruit relationship was a function of physical and biological future conditions as well as random variability (i.e., a climate-informed multispecies model). We fit the model using penalized maximum likelihood to survey biomass, diet, and fishery harvest data for three groundfish species pollock, Pacific cod, and arrowtooth flounder from the EBS in the EBS over the period 1979–2017. We also used the Bering10K ROMSNPZ model to produce detailed hindcasts of temperature for the period 1970–2017. We used hindcast-extracted timeseries from the ROMSNPZ model and CEATTLE model estimates of recruitment ($R_{i,y,l}$) and spawning biomass ($B_{i,y-1}$) in hindcast year $y$ for each species $i$ to fit a climate-enhanced logistic recruitment per

spawner model[36], such that:

$$\ln\left(\hat{R}_{i,y}\right) = \alpha_i - \beta_{0,i}B_{i,y-1} + \ln\left(B_{i,y-1}\right) + \mathbf{B}_i\mathbf{X} + \varepsilon_{i,y} \quad (2)$$

where $\mathbf{B}_i\mathbf{X}_l$ is the summed product of each covariate parameter $\beta_{ij}$ and the corresponding environmental covariate $X_{i,y}$ for each bias-corrected environmental index $j = (1, 2...n_j)$. We selected indices representative of ecological conditions important for groundfish recruitment in the Bering sea[39]; spring and fall large zooplankton abundances, survey replicated bottom temperature, and extent of the residual cold pool of extremely dense and cold sea water that persists across the EBS shelf following spring sea ice retreat. We assumed normally distributed (in log space) residual errors for each species ($\varepsilon_{i,y} \sim N\left(0, \sigma_i^2\right)$). The CEATTLE model was then projected forward where ROMSNPZ indices from individual projections drove growth, predation, and recruitment in each future simulation year[36,62].

**Evaluation of harvest management approaches**. Previous authors have defined EM (i.e., the incorporation of ecosystem information into marine resource management) as a continuum between two paradigms of management and focus[18]. On one end is within-sector single-species management that considers ecosystem information (EAFM) and on the other is cross-sectoral whole of ecosystem management (i.e., EBM). EBFM is intermediate between these and is defined by quantitative incorporation of ecosystem interactions into assessment models and target setting (EBFM). Most fisheries management in the Bering Sea can be characterized as EBFM or EAFM, with increasing trends toward cross-sectoral coordination at the scale of EBM. Here we focus on one aspect on this scale of potential management options, operational EBFM and EAFM as captured through the CEATTLE multispecies stock assessment model and harvest policies made annually under the constraint of the 2 MT cap (modeled via the ATTACH model).

MSE is a process of "assessing the consequences of a range of management strategies or options and presenting the results in a way which lays bare the tradeoffs in performance across a range of management objectives"[63]. MSE has been frequently used to evaluate alternative management strategies based on single-species estimation methods[64]. It is increasingly used to evaluate ecosystem management performance, although these evaluations are far less commonplace due to the complexity of modeling and assessing the performance of ecosystem level metrics[64]. Importantly, MSE "does not seek to proscribe an optimal strategy or decision"[63], rather it aims to describe the uncertainty and tradeoffs inherent in alternative strategies and scenarios. In this case, through a series of workshops, we worked with managers and stakeholders to identify priority scenarios and outputs[19]. From this, risk, sensitivity, and uncertainty under contrasting climate scenarios were requested outputs of the analysis, as was the performance of current climate-naive EBFM policies.

A key component of MSE is identifying and quantifying uncertainty (i.e., process, observation, estimation, model, and implementation error) and representing it using an operating model. In the case of this MSE, the focus was on process error uncertainty due to variation in recruitment about the fitted stock-recruitment relationship, one major source of model error in the form of alternative climate scenarios, and implementation error. The MSE does not account for estimation error (uncertainty in the parameters of the operating model) nor observation error. This is because the estimates of recruitment and spawning biomass from CEATTLE for the BSAI are very precise (see Fig. 10 in ref. [60]) and the estimation and operating models are therefore very similar. Thus, CEATTLE is the operating model for this MSE and implicitly the estimation method. In this approach we assume that while allowing for observation error would have increased overall error, the effect would have been minor compared to the investigated uncertainties. Future analyses using a full MSE (i.e., separate operating and estimation models) could evaluate the effect of observation error, but perhaps more importantly, the potential for model error, whereby the population dynamics model (on which the estimation method is based) differs from that of the operating model such that the estimates on which management decisions are made are biased relative to the true values in the operating model.

Given this we summarized the relative change in catch and biomass for the three species in the model under the following fishing scenarios (Fig. 1): (a) projections without harvest ($F_{i,y} = 0$) in each year $y$ of scenario $l$ for each species $i$, (b) projections under target harvest rate (Supplementary Fig. 7 left) and with a sloping harvest control rule (HCR) (Supplementary Fig. 7 right), (c) as in 2 but with the constraint of a 2 MT cap applied dynamically to the three focal species only.

Under the North Pacific Fishery Management Council (NPFMC) constraint of the 2 MT cap on cumulative total annual catch, realized harvest (i.e., catch) and specification of individual species harvest limits known as Total Allowable Catch (TAC; metric tons) are a function of the acceptable biological catch (ABC) for the given species, as well as ABC of other valuable species in the aggregate complex[19,34] (https://github.com/amandafaig/catchfunction). TAC must be set at or below ABC for each species, therefore TAC of individual species are traded-off with one another to avoid exceeding the 2 MT cap. From 1981 to 1983, the TAC of pollock was reduced significantly below the ABC and in 1984 the 2 MT cap became part of the BSAI fishery management plan[21,34,65]. Pacific cod regulations have changed markedly over recent decades and it was only in the 1990s that in many years the

catch and TAC approached its ABC. Thus, we used the socioeconomic ATTACH model (the R package ATTACHv1.6.0 is available with permission at https://github.com/amandafaig/catchfunction[34]) to model realized catch in each simulation year as a function of CEATTLE assessment estimates of ABC (tons) for pollock, Pacific cod, and arrowtooth flounder under future projections (2018–2100). This entailed three steps for each future simulation year $y$:

1. project the population forward from $y - 1$ to $y$ using estimated parameters from the multispecies mode of the CEATTLE model fit to data from 1979 to 2017 and recruitment based on biomass in simulation year $y$ and future environmental covariates from the ROMSNPZ model downscaled projections (see "Methods" above) to determine $ABC_{i,l}$ for each species ($i$) under each scenario ($l$) given the sloping harvest control rule for pollock, Pacific cod, and arrowtooth flounder in each simulation year $y$;
2. use $ABC_{i,y,l}$ of each species from step 1 as inputs to the ATTACH model in order to determine the North Pacific Marine Fishery Council Total Allowable Catch ($TAC_{i,y,l}$) for the given simulation $l$ year $y$;
3. use $TAC_{i,y,l}$ from step 2 to estimate catch (tons) in the simulation year (Fig. 1); remove catch from the population and advance the simulation forward 1 year.

**Determine the annual ABC.** We used end-of-century projections (2095–2099) to derive a maximum sustainable yield (MSY) proxy for future harvest recommendations ($ABC_{i,y,l}$) for each scenario $l$. To replicate current management, we used a climate-specific harvest control rule that uses climate-naive unfished and target spawning biomass reference points ($B_{0,i}$ and $B_{target,i}$, respectively) and corresponding harvest rates ($F_{i,y} = 0$ and $F_{i,y} = F_{target}$ and $B_{i,y,l}$ in each simulation $l$ year $y$ for each species $i$

$$ABC_{i,y,l} = \sum_a^{A_i} \left( \frac{S_{i,a} F_{ABC,i,y,l}}{Z_{i,a,y,l}} \left(1 - e^{-Z_{i,a,y,l}}\right) N_{i,a,y,l} W_{i,a,y,l} \right) \quad (3)$$

where $W_{i,a,y,l}$, $N_{i,a,y}$, and $Z_{i,a,y,l}$ is the climate-simulation specific annual weight, number, and mortality (i.e., influenced through temperature effects on recruitment, predation, and growth) at age $a$ for $A_i$ ages in the model, and $S_{i,a}$ is the average fishery age selectivity from the estimation period 1979–2017[59,60]. $F_{ABC,i,y,l}$ and was determined in each simulation timestep using an iterative approach[66] whereby we: (i) first determined average $B_{0,i}$ values in years 2095–2099 by projecting the model forward without harvest (i.e., $F_{i,y} = 0$) for each species under the persistence scenario. We then (ii) iteratively solved for the harvest rate that results in an average spawning biomass ($B_{i,y}$) during 2095–2099, that is, 40% of $B_{0,i}$ (i.e., $F_{target,i}$) for pollock and Pacific cod simultaneously, with arrowtooth flounder $F_{i,y}$ set to the historical average (as historical $F$ for arrowtooth flounder $\ll F_{40\%}$); once $F_{target,i}$ for pollock and Pacific cod were found, we then iteratively solved for $F_{target,i}$ for arrowtooth flounder (Supplementary Fig. 7 left panel)[59,60]. Last, (iii) to derive a climate-informed $ABC_{i,y,l}$ in each simulation year, the North Pacific Marine Fisheries Council (hereafter, "Council") Tier 3 sloping harvest control rule with an ecosystem cutoff at $B_{20\%}$ was applied to adjust $F_{ABC,i,y,l}$ lower than $F_{target,i}$ if the simulation specific (climate-informed) $B_{i,y,l}$ was lower than 40% of the climate-naive $B_{0,i}$ at the start of a given year or set to 0 if $B_{i,y,l} < 20\% B_{0,i}$; $F_{ABC,i,y} = F_{target,i,y}$ for the remainder of the simulations where $B_{i,y,l} \geq 40\% B_{0,i}$)[21].

This approach follows the status quo Council reviewed multispecies assessment methodology[60] and represents a precautionary approach that minimizes inflation of ABC due to predator release[59] and also minimizes potential non-intuitive compound effects of climate change and fishing under declining conditions (i.e., whereby a climate-informed $B_0$ declines with climate change and produces a lower target ($B_{40\%}$) biomass). While beyond the scope of this study, future simulations might further explore the performance of alternative $B_0$ approaches (e.g., periodically updated climate-informed $B_0$, annually varying $B_0$ with climate-penalized $B_{40\%}$).

**Simulate TAC.** ATTACH estimates the TAC that would be set by the North Pacific Fishery Management Council, and the subsequent catch that would be harvested by the commercial fishery, based on historical data and assuming current policies and priorities remain relatively unchanged in projections. The impacts of existing policies such as Amendment 80 (i.e., A80, which created multispecies cooperatives), the American Fisheries Act (i.e., AFA, which created pollock cooperatives), and large spatial closures are included and evaluated in the retrospective analysis. Future scenarios will explore relaxation or alteration of these underlying assumptions[19]. ATTACH first estimates the TAC from the specified ABC through an ensemble of three log-linear regressions and seemingly unrelated regressions (SUR) where normally distributed error terms $\varepsilon_{i,y}^{TAC}$ are independent across time, but may have cross-equation contemporaneous correlations[67]. Specifically, the models are statistically fit to historical ABC and TAC data from 1992 to 2017 such

that:

$$\ln\left(TAC_{i,y}\right) = \alpha_i + \beta_i \ln\left(ABC_{i,y}\right) + \sum_{j=1}^{N_j} \gamma_{ij} ABC_{j,y} + \sum_{k=1}^{N_k} \theta_{ik} I_k + \varepsilon_{i,y}^{TAC},$$
$$\text{where } \varepsilon_{i,y}^{TAC} \sim N\left(0, \sigma_i^{TAC}\right) \quad (4)$$

where the harvest limit for species $i$ in a given historical year $y$ ($TAC_{i,y}$) is a function of the assessment model-based ABC ($ABC_{i,y}$, in metric tons) for the species $i$, $\alpha_i$ is the log-linear intercept and $\beta_i$, $\gamma_{ij}$, and $\theta_{ik}$ are coefficients for ABC and policy covariates $I_k$ (e.g., closures, A80, AFA). $\varepsilon_{i,y}^{TAC}$ is the residual error and is log-normally distributed. The residuals of equations estimated as a SUR system are assumed to be correlated, and this is used to more efficiently estimate the regression coefficients. This parameterization assumes exogenous shocks affect all included species to varying degrees. The mean relationship between TAC and ABC (Eq. 4) was used to simulate $TAC_{i,y,l}$ in each projection year using inputs of $ABC_{i,y,l}$ from the CEATTLE model (Eq. 3). Historically, in all but two years the sum of TAC across species has equaled 2 MT exactly, thus we imposed an addition constraint; if the cumulative predicted TACs from the ensemble exceeded 2 MT, the $TAC_{i,y,l}$ of each species was proportionally reduced to satisfy the constraint of a 2 MT limit on the sum of all $TAC_{i,y,l}$. Of note, this step simulates the current management regime, and is not an optimization.

**Simulate annual harvest (catch).** Similarly, ATTACH uses an ensemble of three models to estimate catch biomass ($C_{i,y}$; tons) for a given target species $i$ as a function of $TAC_{i,y}$ and sometimes the $TAC_{j,y}$ of 1 to 2 additional species $j$, as well as relevant policy/events in a given year:

$$\ln\left(C_{i,y}\right) = \alpha_i + \beta_i \ln\left(TAC_{i,y}\right) + \sum_{j=1}^{N_j} \gamma_{ij} TAC_{j,y} + \sum_{k=1}^{N_k} \theta_{ik} I_k + \varepsilon_{i,y}^C,$$
$$\text{where } \varepsilon_{i,y}^C \sim N\left(0, \sigma_i^C\right) \quad (5)$$

The three models in the ensemble differ in their error structure for catch: model 1 assumes each of the log-linear equations are independent, model 2 has two groups of linked SUR (representing species that are caught concurrently), and model 3 has three groups of linked SUR (representing a third group, on top of the two of model 2, of species whose catches are linked). As in (Eq. 4), $\alpha_i$ is the log-linear intercept and $\beta_i$, $\gamma_{ij}$, and $\theta_{ik}$ are coefficients for TAC and policy covariates $I_k$ and $\varepsilon_{i,y}^C$ is the residual error and is log-normally distributed and assumed to be correlated across linked SUR.

The authors of the ATTACH model evaluated the performance of the TAC and catch prediction models using leave-one-out cross-validation (LOO-CV)[34]. For each year, they estimated the coefficients of each model using data from all but that year, and then used those estimators to predict the TAC or catch of the omitted year. They then calculated the difference between the predicted and actual catch for each year (1992–2017) to evaluate the models using a variety of metrics: simple sum of differences, sum of percent differences, sum of squared differences (weighted by value, ABC, TAC, and catch, respectively), the sum of squared percent differences. They also evaluated those metrics for the final models (trained on the entire data set). They found that the ATTACH model performed best for species targeted by directed fisheries, as is the case for the species in this study, while predictive skill of the model was weaker for less economically valuable species. In all species, however, the ATTACH model performed better than assuming catch is equal to ABC (Supplementary Fig. 8).

For the MSE application in this study $ABC_{i,y,l}$ and $ABC_{j,y,l}$ of pollock, Pacific cod, and arrowtooth flounder output from the CEATTLE model were used as inputs into the ATTACH function for each simulation year, while the $ABC_{j,y}$ of the remaining non-CEATTLE species were set to their respective historical averages (1992–2017). Pollock, Pacific cod, and arrowtooth flounder ABCs were used to predict $TAC_{i,y,l}$ for each species, which in turn was used to predict catch $C_{i,y,l}$ using estimated coefficients from the regressions described above (Eq. 4, 5). The $C_{i,y,l}$ of pollock, Pacific cod, and arrowtooth were then removed from the population of each species in the simulation year $y$ (by calculating the effective harvest rate $F_{i,y,l}$ given the input catch $C_{i,y,l}$) and the simulation was rolled forward 1 timestep.

Under this approach, TAC and catch estimates from ATTACH change solely in response to changes in pollock, Pacific cod, and arrowtooth flounder input ABCs. This assumption is significant but difficult to assess without considering alternative models that resolve the population dynamics of the remaining species managed under the 2 MT cap along with their biological interactions and potential future changes in relative harvest values. That said, the three species in CEATTLE are known to strongly interact, account for a large proportion of total EBS fish biomass, and strongly drive TAC allocation. Future sensitivity analyses, possibly using more speciose food web models[32,68] would be useful to quantify this sensitivity.

**Evaluating management performance and risk.** We evaluated the performance of these various management strategies under moderate and high-baseline climate futures. We explored temporal patterns in risk defined as the probability of a 10%

decline in biomass or catch relative to the persistence baseline scenario for each harvest scenario. We also evaluated the probability of severe decline and collapse, defined as a greater than 50 and 80% decline (respectively) in biomass or catch in climate change scenarios (relative to the persistence scenario).

**Threshold and tipping point analysis.** We use ecosystem threshold analyses[69,70] to identify thresholds and tipping points for each species in the non-linear relationship between catch (response) and temperature (pressure). Using the multi-species model estimates of recruitment for each species ($i$) for each future year ($y$) of the simulation and each future scenario ($l$) we drew random samples from the log-normally distributed parameter estimate for climate-enhanced recruitment relationships and projected the model forward under the 2 MT cap management scenario. This formed 100 replicates of each future scenario for each species under no fishing and no cap simulations, and 30 replicates under the 2 MT cap. We calculated the change in catch ($\Delta C_{i,y,l}$) relative to the persistence scenario ($l = 1$) as:

$$\Delta C_{i,y,l} = \left( C_{i,y,l} - C_{i,y,l=1} \right) / C_{i,y,l=1} \tag{6}$$

We then pooled the full set of $\Delta C_{i,y,l}$ and corresponding bottom temperatures ($T$) to estimate the threshold for the $\Delta C_{i,y,l} \sim f(T)$ relationship. Following Samhouri et al.[69], we fit general additive models using thin plate regression spline smoothing terms, using the mgcv[71] package in R, with the basis dimension set to 4 to avoid overfitting. We then calculated the first and second derivative of the smoothing function s(x), as well as the 95% CI (as the 2.75 and 0.975 quantiles) of the smoothing and its second derivative by bootstrapping ($n = 1000$) the residuals. Last, the inflection or tipping point was defined as the temperature whereby the second derivative (s(x)″) changed sign and the 95% CI of the second derivative (smoothed with a local polynomial regression smoother using the loess() function in R with the span set to 10%) was most different from zero[69].

**Reporting summary.** Further information on research design is available in the Nature Research Reporting Summary linked to this article.

## Data availability

All simulation and supporting data for these analyses have been archived and are available for download at https://doi.org/10.6084/m9.figshare.12568625.v1.

## Code availability

Code used to generate intermediate or final data and figures is available for download by Holsman et al.[62]. Ecosystem-based fisheries management forestalls climate-driven collapse analysis and figures code v1.0 (Version v1.0). Zenodo. http://doi.org/10.5281/zenodo.3965248 and https://github.com/kholsman/EBM_Holsman_NatComm. The ATTACH model v1.6.0 and supporting documentation[34] is available for download from the following archived code repository: https://doi.org/10.5281/zenodo.3966545; updated versions can be found at https://github.com/amandafaig/catchfunction.

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

## Acknowledgements

We thank the NOAA's Alaska Climate Integrated Modeling project (ACLIM) team and the FATE project on Climate Impacts on Fish Habitat modeling team for support of this research through collaborative research, development and validation of methodologies, and general conceptualization for integrated modeling of climate change impacts on the Bering Sea ecosystem. Multiple NOAA National Marine Fisheries programs provided support for ACLIM including Fisheries and the Environment (FATE), Stock Assessment Analytical Methods (SAAM) Science and Technology North Pacific Climate Regimes and Ecosystem Productivity, the Integrated Ecosystem Assessment Program (IEA; publication number 2020_5), NOAA Research Transition Acceleration Program (RTAP), the Alaska Fisheries Science Center (ASFC), the Office of Oceanic and Atmospheric Research (OAR) and the National Marine Fisheries Service (NMFS). In addition, the International Council for the Exploration of the Sea (ICES) and the North Pacific Marine Science Organization (PICES) provided support for Strategic Initiative for the Study of Climate Impacts on Marine Ecosystems (SI-CCME) workshops, which facilitated development of the ideas presented in this paper. The scientific views, opinions, and conclusions expressed herein are solely those of the authors and do not represent the views, opinions, or conclusions of NOAA, the Department of Commerce, ICES, or PICES.

## Author contributions

A.B.H., A.H., K.A., A.J.H., and A.E.P. contributed significantly to project conceptualization, funding acquisition, methodology, project administration, and writing of the paper. A.F., A.H., A.J.H., W.C., K.K., and J.I. also contributed to methodology, software development, analysis, writing, and validation of projections and biological modeling. J.C.P.R. and A.E.P. contributed text, analysis, and validation.

## Competing interests

The authors declare no competing interests.
