## [Peer Review File · Nature Communications]

Reviewers' comments, first round

Reviewer #1 (Remarks to the Author):

This paper presents a simulation analysis considering the implications of the utilization of system exploitation caps on the sustainability of currently key target species of fisheries active in the Bering Sea. The paper is well written and laid out and the analyses are all clear and well presented. I do not take issue with the science done, but do seek clarification on a few minor points (see specific comments). I would also suggest a slight tweaking of wording here and there as pointed out in the specific comments, to reflect the fact this is one form of EBM and that a static form was examined rather than a climate aware variant. There is also a slight tension within the paper in that it is discussing EBFM but then only really considers 3 species, key target species, effectively within a classical management arrangement + cap. Despite what the title says it doesn't not give much discussion of broader ecosystem implications or approaches. Now while this is understandable given how hard it is to do the work in the first place and the space limitations of the manuscript format I think a few tweaks here or there could help the manuscript not over sell while still showing the great value of this significant piece of work.

Specific comments

1. Somewhere in the introduction of discussion I would have expected to some discussion of whether the blob (or the period where bottom waters exceeded the threshold in the past, if this was not in the blob period) gave a taste of how accurate the projections are or how suddenly the threshold takes effect on realized fishery outcomes.
2. I think more unpacking of the results would help. Even if kept to supplementary materials. For instance, why is the loss of stabilization faster in RCP 4.5 for Pollock and cod - at least for GFDL and CESM? Also, I'd argue that this also means the stabilization doesn't last through to mid century for pollock in that case, appearing to break down by 2035. Moreover, the cap seems to make the biggest difference for GFDL and almost no difference for MIROC (throughout) and greatest difference for pollock early in RCP 8.5 and late for 4.5 for CESM. What is driving this patterning?
3. Why wasn't the cap calculation repeated mid simulation => to make the cap climate aware? It could be that the approach is still sensible, just that it needs to be made adaptive. It would then allow deeper consideration of economic/social implications. This is no doubt a non-trivial thing to do computationally, so at least comment on this in the discussion in the content of "things to keep in mind". In the discussion you open with "Ecosystem-based management policies" in a broad and sweeping way, but this is a test of one non-adaptive EBFM approach. How did spatial management or other EBFM play in? Where Bo made climate aware in assessment model? Did that help at all? It seems a rather static test as presented. In which case fine first step, but make sure to flag need for follow up given it's says it won't help. Need to point to how it helps beyond the counterfactual of no cap (so classical single spp mgmt.) or indeed no mgmt. at all. Just remember you can't comment on what happens to rest of ecosystem structure and function. This still a very "single" (well target) species presentation and not really reflecting the EBFM vision in reporting.
4. Line 129: I think this would be more appropriately written as "...under the warmest scenarios regardless of the management approaches trialed."
5. Lines 147-152 where zooplankton is discussed. Is it possible to provide some plots of this or how influenced the model content/forcing in the supplementary materials?
6. In the methods please explain (or explain more clearly if you believe you already have covered this) how single species management interacts with the cap. Was pollock or cod always fished near its ABC when "negotiations" on cap happened? How does that F compare to what would have been needed to keep it around?
7. Now as an MSE pedant I would like you to clarify how this is an MSE rather than dynamic simulation testing. I am likely being overly picky (Prof Punt will no doubt will be cringing already), but If CEATTLE was the assessment model, what formed the operating model in the MSE? It seems it simultaneously played both parts - and not in the way where the same model is used separately in each pole rather merged together so there was no true operating model? I think that needs some acknowledgement in the text.

8. Line 234: Where are the $F=0$ projections presented? Perhaps show before/on Figure S1.
9. Line 273: Again this raises the question of a what about climate informed B_0 ?
10. "...each species was proportionally reduced to satisfy the constraint of a 2 MT..." How realistic is this assumption?
11. ATTACH: Are the relationships leading to the good fit for ATTACH up to 2017 likely to continue/change into the future as the system changes?
12. "... while the remaining non-CEATTLE species $ABC_{j,y}$ were set to the historical (1992-2016) average...": Would other ACLIM models agree with this assumption?
13. Figure 3: What causes the regular "humps" in pollock and pacific cod? Why the high upper uncertainty bound on arrowtooth?
14. Figure 5: I think this is a nice presentation
15. Figure 6: What drives non-linearity for arrowtooth or is it that \sim flat and variability appears to give it a non-linear form?

Beth Fulton

Reviewer #2 (Remarks to the Author):

This is an impressive body of work and an important conclusion which I think is well suited for Nature Communications. In particular, I applaud the authors attempts at providing comprehensive source code and data that is required to perform an analysis of this complexity.

The core conclusion of this research is that ecosystem-based management (EBM) approaches may help mitigate some of the impact climate change will likely have on commercially important fish stocks (Pollock and Cod in the Pacific Northwest) relative to non-EBM approaches. I understand that the core element of the EBM approach described here is a the use of a 2 MT annual cap for the harvest of all ground-fish, in addition to setting individual species quotas. My understanding is that the key distinction of such a policy is the application of a cap to ground-fish as a group, rather than traditional non-EBM approaches which consider only set allowable catches species-by-species. I think this makes it a matter of mathematical definition that the total ground-fish catch under an EBM approach in any year is less than or equal to the catch of the non-EBM approach, though I do not believe the authors ever assert that and from the figures shown it is hard for me to directly compare catch in EBM vs non-EBM management line up in a single year. In any event, it seems intuitive that an additional constraint such as the 2 MT cap would on average reduce catch and that would slow declines.

I raise this issue because it is somewhat unclear to what extent the authors assertions about EBM are the result of a detailed, complex modeling and accounting of species/trophic interactions across an ecosystem, and to what extent the result is merely the consequence of any policy, EBM-based or otherwise, that results in a reduction of total catch. It also is not entirely clear what is meant by EBM -- e.g. is any management strategy that sets limits on the sum of across stocks, rather than stocks individually, considered EBM, or does EBM require a model of species/trophic interactions as well?

Beyond my vastly oversimplifying description above, it is quite difficult to assess the details of the analysis presented here owing to the complexity and sheer number of models involved, each with their own assumptions, as beautifully illustrated in Figure 1. Much of this draws on prior art, though I believe few readers of Nature Communications will be familiar with all of the models and acronyms discussed. The management model is presented as an iterative, annual optimization, which I think parallels how most marine fish are actually managed, but it may be worth noting this approach is fundamentally different from the sequential decision (or optimal control) problem for harvesting that is usually considered by the economists and mathematicians. (the sequential decision problem considers all possible sequences of actions over all time, and usually can't be solved in this rich ecosystem context, though see recent results for stage structured fisheries, doi:10.1016/j.mbs.2015.08.021, and see more general theorems in <https://doi.org/10.1007/s00285-018-1275-1>). The authors could do more to spell out the

assumptions of each of the models involved, and the alternatives that exist.

The inclusion of the data and code was a huge help in capturing and communicating the complexity of the various steps, and really for that synthesis alone I would already be happy to see the paper in Nature Comms (though as I have underscored in my opening remarks, I think the overall results are equally compelling and important). However, I must note that I was unable to reproduce the results presented with the code. With a few minor adjustments I can get all but the final supplemental figure to reproduce when running in the "figures only mode". Attempting to generate the results data, SUB_EBM_paper.R gives me the error:

```
```\nread in main dat single spp projections\nProgress: 100% read in main dat multi Progress: 100% reading retro. dataRead 49 items\nRead 137 items\nreading future dataRead 856 items\n[1] "read in projection files"\n\nError in getDat(dat = datIn, scn = ss, val = valIn, sp = spIn, age = age) : object\n'dat_019_CENaivecf_0_5_12_mc101' not found\n```\n
```

In general, despite the evidence of substantial effort from the authors in documenting the bits and pieces, the code and data products could be much better organized to more clearly map to the workflow and models outlined in the paper, more clearly identify what are the "inputs" (existing data and published methods), what are the novel analysis components, and what are the outputs. I realize the immensity of the problem here goes beyond the expectations of publication, so I mention this only as a suggestion. I strongly suggest the authors consider a tool such as `drake`, <https://docs.ropensci.org/drake/>, which is designed precisely to make these kinds of analyses more tractable and reproducible. It may at least be of use to the authors in the future in attempting to effectively package this kind of work.

June 30, 2020  
Reviewer responses to  
“Ecosystem-based fisheries management forestalls climate-driven collapse”  
K. Holsman et al.

**Reviewer comments in bold;** responses not-bolded

---

**Reviewer #1 (Remarks to the Author):**

**This paper presents a simulation analysis considering the implications of the utilization of system exploitation caps on the sustainability of currently key target species of fisheries active in the Bering Sea. The paper is well written and laid out and the analyses are all clear and well presented. I do not take issue with the science done, but do seek clarification on a few minor points (see specific comments).**

**I would also suggest a slight tweaking of wording here and there as pointed out in the specific comments, to reflect the fact this is one form of EBM and that a static form was examined rather than a climate aware variant.**

Thank you for your comment. We have changed EBM to “EBFM” throughout and added text to the introduction to define the scope of these analyses.

**There is also a slight tension within the paper in that it is discussing EBFM but then only really considers 3 species, key target species, effectively within a classical management arrangement + cap. Despite what the title says it doesn't not give much discussion of broader ecosystem implications or approaches.**

We appreciate this comment and would like to clarify that while the assessment model focuses on the 3 target species, the ATTACH model of the 2 MT cap operates across all groundfish in the Fisheries Management Plan to balance biological maximum catch (ABC) with by-catch risk reduction, social and fishery constraints etc. This represents a novel approach to modeling EBFM in the Bering Sea (and elsewhere). We feel that this is consistent with working definitions of EBFM by Dolan et al. 2016 which define EBFM as tools or policies (in this case the 2MT cap) applied across species within the fishery sector, Dolan et al. 2016 state:

“EBFM takes a system-level perspective on fisheries in an ecosystem. Previous works on EBFM have included an exhaustive list of broader goals (e.g. Hall and Mainprize, 2004; Pikitch et al., 2004, 2009; Fletcher, 2005), but a primary take-away from these studies is that managing trade-offs to optimize the overall fisheries yield of an ecosystem over time is the crux of EBFM (Figure 2; Link, 2010). Thus, EBFM differs from EAFM in that it focuses on multiple or all fisheries within an ecosystem and takes a coordinated and strategic approach to providing the greatest benefit to the nation (Patrick and Link, 2015a). Whereas EAFM focuses on a single stock within a fishery and takes a more piecemeal or opportunistic

approach to incorporating ecosystem considerations into management decisions.” (Dolan et al. 2016).

Dolan, T. E., W. S. Patrick, and J. S. Link. 2016. Delineating the continuum of marine ecosystem-based management: A US fisheries reference point perspective. *ICES Journal of Marine Science* 73:1042–1050.

**Now while this is understandable given how hard it is to do the work in the first place and the space limitations of the manuscript format I think a few tweaks here or there could help the manuscript not over sell while still showing the great value of this significant piece of work.**

We agree with this important comment and have changed the text to reflect the title (EBFM) and the fact that we are evaluating EBFM (ecosystem management applied to the fishery sector) rather than EBM (ecosystem management applied across sectors to manage the entire ecosystem). Specifically, we:

- 1) changed “EBM” to “EBFM” throughout the paper to distinguish between EBM (cross sectoral, where the focal aspect of management is the ecosystem) versus EBFM which is multi-species focuses (e.g., accounting for specific trophic interactions) within the context of the ecosystem but does not specifically attempt to manage ecosystem wide objectives.
- 2) added language to the introduction to clarify that the paper considers one type of ecosystem-based management (i.e., is not inclusive of all types of EBM), i.e., that of EBFM (reflecting our title):

*Most of these assessments point to an Ecosystem Management (EM) approach to help promote resilient marine ecosystems and fisheries13,16,17. EM ranges from an Ecosystem Approach to single-species management (i.e., EM as context for management focused on optimizing a single species) to full Ecosystem Based Management (i.e., EM applied across sectors to manage the entire ecosystem). Ecosystem Based Fishery Management (EBFM, i.e., EM applied to the fishery sector) is intermediate to these approaches, and expands classic adaptive management strategies to additionally utilize ecosystem information to manage multiple species across the ecosystem16,18. Intuitively, the more holistic EBFM approach should impart climate-resilience to fisheries, yet few studies have demonstrated the performance of EBFM under climate change (but see17).*

- 3) We added language to the discussion to place the work in the context of what is not evaluated within the paper:

*“We focused our analyses on status quo EBFM policy performance under future conditions, yet these policies were not implemented to specifically address climate change. Our framework enables consideration of a much broader set of scenarios, for instance, implementation of adaptive or climate-informed alternatives to the 2 MT cap, EBM policies that optimize ecosystem productivity (sensu17), or policies that favor climate-informed single species management over current climate-naïve ecosystem caps on TAC. The latter scenario in particular, when considered with the current scenario suite, could*

further characterize the effectiveness of EBFM in facilitating climate adaptation and represent important next steps for evaluation.”

### Specific comments

**1. Somewhere in the introduction of discussion I would have expected to some discussion of whether the blob (or the period where bottom waters exceeded the threshold in the past, if this was not in the blob period) gave a taste of how accurate the projections are or how suddenly the threshold takes effect on realized fishery outcomes.**

We appreciate the reviewer’s suggestion of discussion of recent marine heatwaves in the study region. To add some context of recent marine heatwaves we add the following statement to the introduction (bold and underlined):

*“Indeed, ~~recent and~~ extreme warming and loss of sea ice in the Eastern Bering Sea, especially during recent unprecedented multiyear marine heatwaves in (2014-2019)26, has led to the rapid poleward redistribution of Pacific cod24,25 and declines in recruitment and productivity of several groundfish species26,27.*

However, we did not go into detail regarding the 2014-2016 North Pacific marine heatwave, also known as “the blob”. That heatwave, driven by persistent atmospheric anomalies, primarily impacted the north Pacific and Gulf of Alaska but only extended slightly into the southern Bering Sea. In the northern Bering sea, related but different mechanisms caused anomalous warming in 2016-2019. As is described in Stabeno et al. 2018 in the Bering Sea, warming conditions and marine heatwaves have been observed in recent years, due to large scale changes in winter sea ice extent and arctic warming, which are related but not caused by the same mechanisms leading to the “blob”. The Bering Sea recent marine heatwaves appear to be driven by anomalous winter winds that precluded the establishment of ice in the Northern Bering sea coupled with extremely warm conditions in the Arctic/Chukchi sea and associated lack of winter sea ice. These conditions may be related to long-term anthropogenic driven climate change in combination with internal variability (see conclusions in Walsh et al. 2016) <http://www.ametsoc.net/eee/2016/ch8.pdf> , as is likely the case for the blob as well (warming may contribute to anonymously strong pressure ridges and water-column warming to depth that sustained the blob over multiple seasons).

To address the second portion of this comment (regarding accuracy of projections) we added the following text to the introduction:

*“Observed recent warming, sea ice loss, and biophysical responses in the Bering Sea are consistent with previous projections of impacts of climate change, yet were not anticipated to manifest until midcentury 28. Fishery species have exhibited responses that are both consistent with predictions (e.g., rapid northward distributional shifts of multiple benthic species, declines in recruitment, declines in large lipid-rich zooplankton species)29 and as well as unforeseen responses (e.g., near-term climate resilience of pollock30 or widespread sea bird starvation and mortality events31).”*

**2. I think more unpacking of the results would help. Even if kept to supplementary materials. For instance, why is the loss of stabilization faster in RCP 4.5 for Pollock and cod - at least for GFDL and CESM? Also, I'd argue that this also means the stabilization doesn't last through to mid century for**

**pollock in that case, appearing to break down by 2035. Moreover, the cap seems to make the biggest difference for GFDL and almost no difference for MIROC (throughout) and greatest difference for pollock early in RCP 8.5 and late for 4.5 for CESM. What is driving this patterning?**

We appreciate the reviewer's perspective here and agree the complexity of results makes it difficult to initially "see" the drivers of the response. A few points of clarification:

**why is the loss of stabilization faster in RCP 4.5 for Pollock and cod - at least for GFDL and CESM?**

We think there may be some slight confusion with Figures 2, 4, and S2, we have adjusted the colors slightly for GFDL 4.5 to help distinguish it from MIROC and help clarify the pattern (we have also screened these using the program Sim Daltonism). First, the loss of stabilization does not occur faster for GFDL simulations under RCP 4.5, --i.e., under the 2 MT cap and GFDL RCP 4.5, catch *does not decline appreciably* from the persistence scenario baseline for Pacific cod, and is only slightly reduced for pollock, a trend observed in both cap and no cap scenarios. However, loss of stabilization does occur rapidly under MIROC (warmest during 2025-2040) and CESM (also very warm relative to GFDL but slightly cooler than MIROC during 2025-2040; Fig. 2). Similarly, MIROC RCP 8.5 is extremely warm during this early period in the projections and is associated with a marked decline in stabilization for pollock ~2040. The effect of the cap (i.e., catch in simulations with the 2 MT cap > catch in simulations without the 2 MT cap effects) is greatest for arrowtooth, followed by pollock, and least for Pacific cod. For these reasons we conclude that the response reflects a combination of thermal thresholds (Fig6) and effective harvest rate under the 2 MT cap (FigS3). Our main takeaway is that EBFM can forestall collapse under warming conditions, and this pattern scales with temperature, in that the faster the system warms the sooner catch will decline non-linearly even under EBFM. We discuss this in the second paragraph of the discussion. To address the comment without adding significant length to the text we additionally clarified the first and last sentences of the second paragraph of the discussion (new text underlined):

*"EBFM policies interact with climate sensitivity to impact species differently. In this example, the EBFM 2 MT cap tips the balance for arrowtooth flounder such that future catch is stable and sometimes increases under future climate change. Yet for Pacific cod, the 2 MT cap EBFM policy has little effect on catch and therefore does not alter the outcome of declines under climate change. This reflects historical precedent of the effect of the 2 MT cap for maximizing catch of the most valuable species in the aggregate complex (i.e., Pacific cod)29. Contrasting effects of the 2 MT cap across species suggest that EBFM harvest policies that reduce the spawning exploitation rate at high abundances should buffer stocks30 and reduce vulnerability to climate-induced collapse, at least in the near-term. Of note, the benefit of the EBFM 2 MT cap scales with rate of warming, with collapse occurring before 2050 in the fastest warming simulations even under the 2 MT cap."*

We also added text to the results to clarify that reduction in the benefits of the cap were linked to the rate of warming (end of paragraph 3 of the results).

**? Also, I'd argue that this also means the stabilization doesn't last through to mid century for pollock in that case, appearing to break down by 2035.**

We agree and would like to point the reviewer to the third paragraph of the discussion where we state: *“Prior to 2050, the general trend is a decline relative to the persistence scenario, but there is considerable variability among GCMs due to process error.”*

**Moreover, the cap seems to make the biggest difference for GFDL and almost no difference for MIROC (throughout) and greatest difference for pollock early in RCP 8.5 and late for 4.5 for CESM. What is driving this patterning?**

This reflects that the cap has a stabilizing effect for pollock up until bottom temperatures pass ~2.1 deg C, after which point declines occur in both “cap” and “no cap” simulations if the RCPxGCM has crossed 2.1deg C in bottom temperature. This is discussed in the fourth paragraph of the discussion.

**3. Why wasn't the cap calculation repeated mid simulation => to make the cap climate aware? It could be that the approach is still sensible, just that it needs to be made adaptive. It would then allow deeper consideration of economic/social implications. This is no doubt a non-trivial thing to do computationally, so at least comment on this in the discussion in the content of “things to keep in mind”. In the discussion you open with “Ecosystem-based management policies” in a broad and sweeping way, but this is a test of one non-adaptive EBFM approach. How did spatial management or other EBFM play in? Where Bo made climate aware in assessment model? Did that help at all? It seems a rather static test as presented. In which case fine first step, but make sure to flag need for follow up given it's says it won't help. Need to point to how it helps beyond the counterfactual of no cap (so classical single spp mgmt.) or indeed no mgmt. at all. Just remember you can't comment on what happens to rest of ecosystem structure and function. This still a very "single" (well target) species presentation and not really reflecting the EBFM vision in reporting.**

There are a few specific points made in this comment which we group into two topics:

[1. EBM vs EBFM clarifications/discussion] **In the discussion you open with “Ecosystem-based management policies” in a broad and sweeping way, but this is a test of one non-adaptive EBFM approach. How did spatial management or other EBFM play in? ... Just remember you can't comment on what happens to rest of ecosystem structure and function. This still a very "single" (well target) species presentation and not really reflecting the EBFM vision in reporting.**

Please see discussion of our steps to address this issue in the first response to EBM/EBFM above. To recap, we generally agree with the reviewer and have clarified the text to reflect that we have adopted the Dolan et al. 2016 definition of Ecosystem Based Fisheries Management (EBFM) which includes multispecies management and quantitative accounting of environmental and/or trophic interactions in the management of a few target species. This differs but often complements EBM which is a primarily place-based cross sectoral management of the ecosystem as a whole, with targets such as optimizing ecosystem productivity, trophic balance etc. Under this, we maintain that the focus of this paper is on current status quo management performance under climate change. Status quo management in the EBS includes many ecosystem considerations and EBFM measures such as the B20% cutoff in harvest (FigS1) and the 2 MT cap, which interacts with bycatch restrictions on non-target species, species of concern, and spatial-time management to prevent bycatch (all modeled through the ATTACH model).

In addition to the text in the introduction and discussion we added the following text to the methods:

*“Previous authors have defined Ecosystem Management (i.e., the incorporation of ecosystem information into marine resource management) as a continuum between two paradigms of management and focus 18. On one end is within-sector single-species or single habitat management that considers ecosystem information (EAM) and on the other is cross-sectoral whole of ecosystem management (i.e., EBM). Ecosystem Based Fisheries Management (EBFM) is intermediate between the two, distinguished from one another through quantitative incorporation of ecosystem interactions into assessment models and target setting (EBFM) and through ecosystem context for harvest policies (EAFM). Most fisheries management in the Bering Sea can be characterized as EBFM or EAFM, with increasing trends towards cross-sectoral coordination at the scale of EBM. Here we focus on one aspect on this scale of potential management options, operational EBFM and EAFM as captured through the CEATTLE multispecies stock assessment model and harvest policies decisions made annually under the constraint of the 2 MT cap (modeled via the ATTACH model).”*

**[2. Testing of climate informed tools] Why wasn't the cap calculation repeated mid simulation => to make the cap climate aware? It could be that the approach is still sensible, just that it needs to be made adaptive. It would then allow deeper consideration of economic/social implications. This is no doubt a non-trivial thing to do computationally, so at least comment on this in the discussion in the content of “things to keep in mind”....**

We are excited that the reviewer sees potential future applications of the framework introduced in this paper. We agree that these alternative methods represent climate-informed approaches of interest and would add dynamic elements to current, static management approaches. However, it represents additional computational and scenario analyses beyond the scope of this paper. Evaluating climate-informed policy and assessment tools is the focus of ongoing work and is complex requiring, in our opinion, considerable consultation and cooperation with managers and stakeholders. As such we are in the process of co-developing sets of priority simulations and approaches to evaluate going forward, including evaluation of climate informed harvest control rules and management policies. This is currently underway in partnership with the North Pacific Marine Fisheries Council with significant input from stakeholders and managers. We have not yet fully evaluated the suite of potential policies as they are still evolving with ongoing discussions of our emergent ACLIM results.

However, this paper outlines important first results from the key research priority identified by stakeholders and managers during scoping workshops in 2016-2018, i.e., it specifically aims to address whether present-day (status quo) EBFM and EAFM measures can be used to address climate-driven change. Results presented herein have informed the “climate-ready” evaluations that are presently underway and will be synthesized in the next few years through ACLIM (see Hollowed et al. 2020 for a discussion of the scope of those planned evaluations). Also, of note, the current 2 MT cap is intentionally designed to be invariable in time or biomass, a feature that can impart stability during changing political/social priorities (see Holsman et al. 2019. Towards climate resiliency in fisheries management. ICES Journal of Marine Science 76:1368–1378). Evaluation of cap alternatives is being conducted with close input and discussion from the Council, as modifying the cap through legislative processes is an in-depth political discussion that mandates a cautious approach and collaborative determination of measures of performance that reflect diverse stakeholder perspectives.

Given this we appreciate the comment and have added the following text to the end of the discussion:

*“We focused our analyses on status quo EBFM policy performance under future conditions, yet these polices were not implemented to specifically address climate change. Our framework enables*

*consideration of a much broader set of scenarios, for instance, implementation of adaptive or climate-informed alternatives to the 2 MT cap, EBM policies that optimize ecosystem productivity (sensu 17), or policies that favor climate-informed single species management over current climate-naïve ecosystem caps on TAC. . When considered with the current scenario suite, such scenarios could further characterize the effectiveness of EBFM in facilitating climate adaptation and represent important next steps for evaluation.”*

**Where  $B_0$  made climate aware in assessment model? Did that help at all? It seems a rather static test as presented. In which case fine first step, but make sure to flag need for follow up given it's says it won't help. Need to point to how it helps beyond the counterfactual of no cap (so classical single spp mgmt.) or indeed no mgmt. at all.**

We used a climate naïve  $B_0$  and a climate-informed  $B_{40}$  based on prior evaluations that showed this approach performs better under unidirectional climate change than either climate naïve  $B_0$  and  $B_{40}$  or climate informed  $B_0$  and  $B_{40}$ . Specifically, a climate naïve  $B_0$  used in combination with climate-informed  $B_{40}$  targets performed more favorably in previous exploration in terms of allowing catch while forestalling collapse. This methodology is what is presently used in the operational CEATTLE assessment (annually since 2017) (Holsman et al. 2019,2018, 2017 as appendices to the BSAI pollock stock assessment Iannelli et al. 2019) links here:

<https://archive.afsc.noaa.gov/refm/docs/2019/EBSmultispp.pdf> and here:

<https://www.fisheries.noaa.gov/alaska/population-assessments/north-pacific-groundfish-stock-assessments-and-fishery-evaluation>.

To clarify this we modified the following text in the methods to read:

*“To replicate current management strategies we used a climate-specific harvest control rule that uses a climate-naïve unfished ( $F = 0$ ) spawning biomass target ( $B_0$ ) and a climate-informed target spawning biomass ( $B_X\%$ ), i.e. a precautionary approach 40,63. This follows the status quo Council reviewed multispecies assessment methodology that minimizes potential non-intuitive compound effects of climate change and fishing under declining conditions (i.e., whereby a climate informed  $B_0$  declines with climate-change and produces a lower target ( $B_X\%$ ) biomass).”*

Finally, as suggested in the methods we flagged future directions that build upon this work:

*“While beyond the scope of this study, future simulations might further explore the performance of alternative  $B_0$  approaches (e.g., periodically updated climate-informed  $B_0$ , annually varying  $B_0$  with climate-penalized  $B_X\%$ ).”*

Other considerations that are relevant to this comment:

- We also agree that dynamic versus static  $B_0$  is an interesting topic that is beyond the scope of this study (which aims to evaluate status quo ecosystem-based policies under climate change). Dynamic  $B_0$  (and other climate-informed reference point approaches) are part of an ongoing evaluation in ACLIM (see Hollowed et al. 2020).
- We used static  $B_0$  for this paper following the status quo Council reviewed multispecies assessment approach whereby exploratory simulations indicated that dynamic  $B_0$  can have non-intuitive compounding effects with climate change under declining conditions,-- i.e., a declining

B0 with climate-change results in less conservative and lower target biomass reference points (as is now mentioned in the methods in more detail). To help illustrate this we have plotted projections of catch with ABC from climate informed B0 and climate naive B0 below, both run through the ATTACH model (Fig. X1).

Fig. X1. Comparative plots of projections of catch over time given climate-informed B0 and climate informed B40 (left hand column) or climate naive B0 and climate informed B40 (right hand column; this paper). Climate informed B0 (lefthand column) results in ABC = 0, because unfished SSB ( $F=0$ ) declines to near zero in RCP 8.5 simulations.

**4. Line 129: I think this would be more appropriately written as “...under the warmest scenarios regardless of the management approaches trialed.”**

We agree and made the suggested change.

**5. Lines 147-152 where zooplankton is discussed. Is it possible to provide some plots of this or how influenced the model content/forcing in the supplementary materials?**

We added a plot of fall zooplankton to the supplemental material (now Figure S7). These indices can also be explored through an interactive shiny() tool online: <https://kholzman.shinyapps.io/aclim>

Detailed discussion of the inverse relationship of fall zooplankton and temperature trends can be found in Hermann et al. 2013,2019 (and are cited in the text).

**6. In the methods please explain (or explain more clearly if you believe you already have covered this) how single species management interacts with the cap. Was pollock or cod always fished near its ABC**

**when "negotiations" on cap happened? How does that F compare to what would have been needed to keep it around?**

As requested, additional text was added in the methods section to better explain the history of the 2 MT cap and its relationship to the single-species TAC-setting and fishery actions, this text now reads:

*“Under the North Pacific Fishery Management Council (NPFMC) constraint of the 2 MT cap on cumulative total annual catch, realized harvest (“catch”) and specification of individual species harvest limits known as TAC (“Total Allowable Catch”; metric tons) are a function of the Acceptable Biological Catch (ABC) for the given species, as well as ABC of other valuable species in the aggregate complex 19,33. TAC must be set at or below ABC for each species, therefore TAC of individual species are traded-off with one another to avoid exceeding the 2 MT cap. From 1981-1983, the TAC of pollock was reduced significantly below the ABC and in 1984 the 2 MT cap became part of the BSAI fishery management plan 21,33,64. Pacific cod regulations have changed markedly over recent decades and it was only in the 1990s that in many years the catch and TAC approached its ABC. Thus, we used the socioeconomic ATTACH model (the R package ATTACHv1.1.0 is available with permission at <https://github.com/amandafaig/catchfunction> 33) to model “realized catch” in each year simulation year as a function of CEATTLE assessment estimates of Acceptable Biological Catch (ABC; tons) for pollock, Pacific cod, and arrowtooth flounder in a given simulation year from 2017-2100. “*

As we note at different points in the text, management of both pollock and cod have changed over the years and has impacted the ABC-TAC-catch relationship. During the study period, we are able to predict the out-of-sample TAC-setting and catch with high accuracy given the many objectives of the policy makers. We also added text in the introduction that better describes how the fishery management Council reduces TACs for different species to meet diverse management objectives and mandates. At its lowest historical levels, the pollock TAC has been equal to its ABC, as in that case there is enough space under the cap to catch the TAC at that level while ensuring fishing opportunities for other species. In the introduction we added the following text to help clarify the role of the cap (end of paragraph 3):

*“Managers reduce harvest limits for individual stocks to conform to the 2 MT cap based on multiple management objectives, including maximizing sustainable yield, reducing the risk of exceeding directed and incidental catch limits (which can close a fishery for the season), other ecosystem considerations and impacts, and meeting distributional objectives and mandates21..”*

**7. Now as an MSE pedant I would like you to clarify how this is an MSE rather than dynamic simulation testing. I am likely being overly picky (Prof Punt will no doubt will be cringing already), but If CEATTLE was the assessment model, what formed the operating model in the MSE? It seems it simultaneously played both parts – and not in the way where the same model is used separately in each pole rather merged together so there was no true operating model? I think that needs some acknowledgement in the text.**

The reviewer has characterized our methodology correctly. A key component of Management Strategy Evaluation is identifying and quantifying uncertainty, which is usually divided into process, observation, estimation, model and implementation error, and representing it in an operating model. A primary aim of an MSE is to compare management scenarios in collaboration with managers, such that outputs of analyses can provide actionable advice (Smith, 1994). It is computationally infeasible to consider all possible uncertainties so all MSE studies involve selecting the key subset of uncertainties (Punt et al.,

2016). In the case of this MSE, the focus was on process error uncertainty due to variation in recruitment about the fitted stock-recruitment relationship, one major source of model error in the form of alternative climate scenarios, and implementation error. The MSE does not account for estimation error (uncertainty in the parameters of the operating model) nor observation error. This is because the estimates of recruitment and spawning biomass are very precise (see Figure 10 of Holsman et al. (2019)). Allowing for observation error would have increased overall error but without expanding the study substantially by including model error (i.e. the estimation method is not the same as the operating model) the effect would have been minor compared to the investigated uncertainties

We deeply respect the reviewer's expertise in this matter and welcome further discussion. Towards this we added the following text to the methods:

*“Management Strategy Evaluation (MSE) is a process of “assessing the consequences of a range of management strategies or options and presenting the results in a way which lays bare the tradeoffs in performance across a range of management objectives” (Smith 1994). MSE has been frequently used to evaluate alternative management strategies based on single-species estimation methods (Punt et al. 2016). It is increasingly used to evaluate ecosystem management performance, although these evaluations are far less commonplace due to the complexity of modeling and assessing the performance of ecosystem level metrics (Punt et al. 2016). Importantly, MSE “does not seek to proscribe an optimal strategy or decision” Smith 1994, rather it aims to describe the uncertainty and tradeoffs inherent in alternative strategies and scenarios. As part of the ACLIM project, we held a series of workshops with managers and stakeholders to develop scenarios for evaluation and to identify performance criteria (where the methodology should be designed to best address these questions/scenarios). In this case, through a series of workshops we worked with managers and stakeholders to identify priority scenarios and outputs (Hollowed et al. 2020). From this risk sensitivity and uncertainty were requested outputs of the analysis, as was the performance of climate-naïve EBFM policies currently in place.*

*A key component of Management Strategy Evaluation is identifying and quantifying uncertainty (i.e., process, observation, estimation, model and implementation error) and representing it using an operating model. In the case of this MSE, the focus was on process error uncertainty due to variation in recruitment about the fitted stock-recruitment relationship, one major source of model error in the form of alternative climate scenarios, and implementation error. The MSE does not account for estimation error (uncertainty in the parameters of the operating model) nor observation error. This is because the estimates of recruitment and spawning biomass from CEATTLE for the BSAI are very precise (see Figure 10 of Holsman et al. (2019)). Allowing for observation error would have increased overall error but without expanding the study substantially by including model error (i.e. the estimation method is not the same as the operating model) the effect would have minor compared to the investigated uncertainties. Thus, CEATTLE is the operating model for this MSE and implicitly the estimation method.”*

That said, if the reviewer still feels strongly after this explanation that “dynamic simulation testing” should be used in place of “MSE” we will replace “MSE” through-out the paper.

**8. Line 234: Where are the F=0 projections presented? Perhaps show before/on Figure S1.**

Figure 3 shows the F=0 (unfished biomass) projections. We added “F=0” to the methods to clarify that unfished biomass refers to F=0.

### 9. Line 273: Again this raises the question of a what about climate informed Bo?

See comments above. Again (as mentioned above) the following text was added to the methods:

*“To replicate current management strategies we used a climate-specific harvest control rule that uses a climate-naïve unfished ( $F = 0$ ) spawning biomass target ( $B_0$ ) and a climate-informed target spawning biomass ( $BX\%$ ), i.e. a precautionary approach 40,63. This follows the status quo Council reviewed multispecies assessment methodology that minimizes potential non-intuitive compound effects of climate change and fishing under declining conditions (i.e., whereby a climate informed  $B_0$  declines with climate-change and produces a lower target ( $BX\%$ ) biomass). While beyond the scope of this study, future simulations might further explore the performance of alternative  $B_0$  approaches (e.g., periodically updated climate-informed  $B_0$ , annually varying  $B_0$  with climate-penalized  $BX\%$ ).”*

### 10. “...each species was proportionally reduced to satisfy the constraint of a 2 MT...” How realistic is this assumption?

When estimating the TAC of each species from its ABC with ATTACH, we found better performance from models that did not include all species as covariates for each species. While this usually does not lead to a net TAC surpassing 2 MT, it raises the possibility that small adjustments are needed to avoid exceeding the cap. When the predicted TAC for all species does exceed 2 MT we felt reducing the TAC estimate for most species by the same proportion was the best solution as it would spread the impact of this reduction so that the most abundant species (pollock, Pacific Cod, and yellowfin sole) experienced the largest nominal decrease in TAC. These species are not only significantly more abundant than the rest, but they are also the most likely to cause the predicted net TAC to exceed 2 MT. (“Most species” excludes sablefish, Shortraker rockfish, and Northern rockfish, as these species’ TACs have such low variance that a reduction would necessarily worsen the species-specific and aggregate predictions.)

### 11. ATTACH: Are the relationships leading to the good fit for ATTACH up to 2017 likely to continue/change into the future as the system changes?

ATTACH is fit to historical data, a time period when there has been considerable variation in ABCs, evolving assessment methods, and new management measures. Under those conditions, the regressions developed predict out-of-sample surprisingly well. However, it is likely that as the system moves to previously unseen combinations of abundances of different species, management may change in ways historical analysis and ATTACH cannot anticipate. For example, if pollock and cod became a significantly smaller portion of the ecosystem than historically observed, the system would likely prioritize greater exploitation of flatfish under the 2 MT cap. While ‘status quo’ inherently makes assumptions that management does not change beyond the observed range, it further assumes that management in previously un-encountered situations would be consistent with historical management in how ABCs are transformed to TACs and in the share of each TAC that the fishery is able to catch. This most likely would mean status quo would deviate from previously observed levels, as ATTACH does not adapt to novel situations or changing objectives in the manner that the fishery management council is able to do so. This represents an important limitation of using retrospective patterns of social-ecological-climate coupling to project future conditions and is why we place emphasis on performance and relative risk across scenarios rather than predictive skill. It is our hope that these analyses identify the strengths and short-comings of current climate-naïve status quo approaches, highlighting the need to further develop and evaluate climate-ready policies and tools in future analyses.

Towards this we added the following text to the methods to highlight these assumptions in policy stationarity:

*“ATTACH estimates the Total Allowable Catch (TAC) that would be set by the North Pacific Fishery Management Council, and the subsequent catch that would be harvested by the commercial fishery, based on historical data and assuming current policies and priorities remain relatively unchanged in projections. The impacts of existing policies such as Amendment 80 (“A80”, which created multispecies cooperatives), the American Fisheries Act (“AFA”, which created pollock cooperatives), and large spatial closures are included and evaluated in the retrospective analysis. Future scenarios will explore relaxation or alteration of these underlying assumptions19”*

#### **12. “... while the remaining non-CEATTLE species ABCj,y were set to the historical (1992-2016) average...”: Would other ACLIM models agree with this assumption?**

This sentence refers to ATTACH inputs at each time-step (i.e., ABC); we would like to note that outputs of ATTACH (i.e., TAC and catch) of other non-CEATTLE species may change based on the annual pollock, cod, and arrowtooth changes in ABC. That said, the inputs of ABC for non-CEATTLE species are set to average values for simplicity and ease of interpretation of results (3 GCMs x 2 RCPs X 3 harvest/management strategies is already a large suite of simulations to evaluate). Other models within the ACLIM suite include non-CEATTLE species and do simulate dynamic ABC inputs across species (e.g. see Hollowed et al. 2020). For example, Reum et al. 2020 coupled a size-spectrum model to ATTACH and found three “clades” of species response (their Fig 3); pollock and cod (and various other species) were grouped according to dominance by internal sources of uncertainty, whereas arrowtooth (and flathead sole) were more sensitive to uncertainty in fishing scenarios. Synthesis of these results across the ACLIM suite of biological models is forthcoming and will further elucidate the sensitivity of this assumption. From that, additional simulations using a range of ABCs for other species, for example, may be used to simulate future conditions (Hollowed et al. 2020)

Reum, J. C. P., J. L. Blanchard, K. K. Holsman, K. Aydin, A. B. Hollowed, A. J. Hermann, W. Cheng, A. Faig, A. C. Haynie, and A. E. Punt. 2020. Ensemble Projections of Future Climate Change Impacts on the Eastern Bering Sea Food Web Using a Multispecies Size Spectrum Model. *Frontiers in Marine Science* 7:1–17.

#### **13. Figure 3: What causes the regular "humps" in pollock and pacific cod? Why the high upper uncertainty bound on arrowtooth?**

The “humps” are due to climate-driven pulses in recruitment (or periodic recruitment failures) and predation (by species in the model and operating mostly on ages 1 and 2) which manifest across age classes and years (maturation of strong year classes) and are evident by comparing climate change scenarios to the persistence scenario (constant climate, but same SR relationship). Climate and density dependence effects on recruitment are less pronounced (lower parameter value) for arrowtooth and thus do not induce as large of oscillations.

For the same reasons (lower parameter values and more residual error) there is higher uncertainty around the SR with climate relationship for arrowtooth, resulting in larger quantiles during the random draws from the recruitment estimates in future projections (tighter bounds in the other species are due to tighter relationships between recruitment, spawning biomass and environmental covariates). To clarify this the legend was modified to include :

*“Shading indicates the 10th and 90th quantiles from 100 random draws from estimated recruitment parameters.”*

**14. Figure 5: I think this is a nice presentation**

We deeply appreciate the reviewer’s comment.

**15. Figure 6: What drives non-linearity for arrowtooth or is it that ~flat and variability appears to give it a non-linear form?**

**Beth Fulton**

It is non-linear but according to the definition of tipping point and threshold from Samhouri et al. 2017 the non-linearity is not significant (bounds of the first derivative overlap 0). This indicates that arrowtooth catch does not exhibit a significant threshold effect or tipping point relative to temperature (although it does potentially decline non-linearly without the cap in place).

---

**Reviewer #2 (Remarks to the Author):**

This is an impressive body of work and an important conclusion which I think is well suited for Nature Communications. In particular, I applaud the authors attempts at providing comprehensive source code and data that is required to perform an analysis of this complexity.

The core conclusion of this research is that ecosystem-based management (EBM) approaches may help mitigate some of the impact climate change will likely have on commercially important fish stocks (Pollock and Cod in the Pacific Northwest) relative to non-EBM approaches. I understand that the core element of the EBM approach described here is the use of a 2 MT annual cap for the harvest of all ground-fish, in addition to setting individual species quotas. My understanding is that the key distinction of such a policy is the application of a cap to ground-fish as a group, rather than traditional non-EBM approaches which consider only set allowable catches species-by-species.

Yes this is correct.

I think this makes it a matter of mathematical definition that the total ground-fish catch under an EBM approach in any year is less than or equal to the catch of the non-EBM approach, though I do not believe the authors ever assert that and from the figures shown it is hard for me to directly compare catch in EBM vs non-EBM management line up in a single year. In any event, it seems intuitive that an additional constraint such as the 2 MT cap would on average reduce catch and that would slow declines.

We appreciate the reviewer's comment and would like to highlight Figure 4 which is the "side by side" change in catch (relative to persistence scenario) for EBFM ("2 MT cap") and non-EBFM ("no cap") simulations.

We would also like to point the reviewer to the following section in the methods (there is no associated figure but the text does describe the aggregate cap) under step 3 in the methods (bottom of page 16, top of page 17):

*"Historically, in all but two years the sum of TAC across species has equaled 2 MT exactly, thus we imposed an addition constraint; if the cumulative predicted TACs from the ensemble exceeded 2 MT, the  $TAC_{i,y}$  of each species was proportionally reduced to satisfy the constraint of a 2 MT limit on the sum of all  $TAC_{i,y}$ ."*

I raise this issue because it is somewhat unclear to what extent the authors assertions about EBM are the result of a detailed, complex modeling and accounting of species/trophic interactions across an ecosystem, and to what extent the result is merely the consequence of any policy, EBM-based or otherwise, that results in a reduction of total catch. It also is not entirely clear what is meant by EBM - e.g. is any management strategy that sets limits on the sum of across stocks, rather than stocks individually, considered EBM, or does EBM require a model of species/trophic interactions as well?

We appreciate the reviewer's comment and would like to clarify our perspective of what delineates Ecosystem Based Fisheries Management (EBFM) from non-EBM/EBFM management. It is our opinion

and experience that management approaches are defined through their objectives, implicit versus explicit assumptions, and goals, rather than the specific tools implemented to achieve those goals, -- i.e., what distinguishes EBFM from single species management is the differing suite of objectives of each approach (e.g., maximize productivity across species in EBFM vs maximize individual species productivity in single species management), not so much the tool(s) used to achieve those objectives (i.e., reduce annual harvest). Ecosystem based fishery management (EBFM), like any management approach, can use the same or different set of tools ( such as time-area closures, species and by-catch restrictions, etc. ) and these can be, and are often, the same in EBFM and non-EBFM. However, it is the objectives and intention of the tools that are the basis for EBFM, --i.e., it is the fact that the cap is implemented to address ecosystem productivity and multispecies objectives that render the cap as a tool for EBFM (see Dolan et al. 2016 for more on this). Similarly, the minimum biomass threshold of 20% is designed to protect food resources for marine mammals and other species in the ecosystem but it acts through reducing  $F$  to 0 when  $B < B_{20\%}$ . We agree the effect of the cap could be replicated through many approaches and might represent a novel method of implementing EBFM-like effects in single species management in future analyses. Similarly, here we aimed to evaluate if these measures implemented to address ecosystem-level objectives also incur climate resilience. That said, it is important to note that the emergent patterns in effective  $F$  (FigS3) from ATTACH are outcomes of the implicit EBFM policies and tradeoff evaluations done annually given the 2 MT cap constraints over the recent history.

The reviewer does however point the need to define EBFM in our paper. To address this we added the following text to the methods (as well as discussion of EBFM to the discussion and introduction):

*“Previous authors have defined Ecosystem Management (i.e., the incorporation of ecosystem information into marine resource management) as a continuum between two paradigms of management and focus 18. On one end is within-sector single-species management that considers ecosystem information (EAFM) and on the other is cross-sectoral whole of ecosystem management (i.e., EBM). Ecosystem Based Fisheries Management (EBFM) is intermediate between these and is defined by quantitative incorporation of ecosystem interactions into assessment models and target setting (EBFM). Most fisheries management in the Bering Sea can be characterized as EBFM or EAFM, with increasing trends towards cross-sectoral coordination at the scale of EBM. Here we focus on one aspect on this scale of potential management options, operational EBFM and EAFM as captured through the CEATTLE multispecies stock assessment model and harvest policies decisions made annually under the constraint of the 2 MT cap (modeled via the ATTACH model).*

**Beyond my vastly oversimplifying description above, it is quite difficult to assess the details of the analysis presented here owing to the complexity and sheer number of models involved, each with their own assumptions, as beautifully illustrated in Figure 1. Much of this draws on prior art, though I believe few readers of Nature Communications will be familiar with all of the models and acronyms discussed. The management model is presented as an iterative, annual optimization, which I think parallels how most marine fish are actually managed, but it may be worth noting this approach is fundamentally different from the sequential decision (or optimal control) problem for harvesting that is usually considered by the economists and mathematicians. (the sequential decision problem considers all possible sequences of actions over all time, and usually can't be solved in this rich ecosystem context, though see recent results for stage structured fisheries, doi:10.1016/j.mbs.2015.08.021, and see more general theorems in <https://doi.org/10.1007/s00285-018-1275-1>). The authors could do more to spell out the assumptions of each of the models involved, and the alternatives that exist.**

We would like to first clarify that the management (and commercial harvest) ATTACH model is an annual prediction based on data-based regressions, rather than an optimization (consistent with an MSE approach, Smith 1994). The prediction is based on management and commercial harvest as it has historically been executed, and is intended to predict how the system would behave in the context of evolving ABCs based on “status quo” management policies.

We have made the following edit to the manuscript, to better clarify what ATTACH (i.e., the management and commercial harvest model) does, including that it does not optimize dynamically or annually.

At the beginning of “Step 2” of the section “Evaluation of harvest management approaches” in the (online) Materials and Methods we have added the following description:

*“ATTACH estimates the Total Allowable Catch (TAC) that would be set by the North Pacific Fishery Management Council, and the subsequent catch that would be harvested by the commercial fishery, based on historical data and assuming current policies and priorities remain relatively unchanged in projections. The impacts of existing policies such as Amendment 80 (“A80”, which created multispecies cooperatives), the American Fisheries Act (“AFA”, which created pollock cooperatives), and large spatial closures are included and evaluated in the retrospective analysis. Future scenarios will explore relaxation or alteration of these underlying assumptions19”*

And at the end of this section we have added:

*“Of note, this step simulates the current management regime, and is not an optimization.”*

The inclusion of the data and code was a huge help in capturing and communicating the complexity of the various steps, and really for that synthesis alone I would already be happy to see the paper in Nature Comms (though as I have underscored in my opening remarks, I think the overall results are equally compelling and important). However, I must note that I was unable to reproduce the results presented with the code. With a few minor adjustments I can get all but the final supplemental figure to reproduce when running in the “figures only mode”. Attempting to generate the results data, SUB\_EBM\_paper.R gives me the error:

```
...
read in main dat single spp projections
Progress: 100% read in main dat multi Progress: 100% reading retro. dataRead 49 items
Read 137 items
reading future dataRead 856 items
[1] "read in projection files"

Error in getDat(dat = datIn, scn = ss, val = valIn, sp = spln, age = age) : object
'dat_019_CENaivecf_0_5_12_mc101' not found
...
```

In general, despite the evidence of substantial effort from the authors in documenting the bits and pieces, the code and data products could be much better organized to more clearly map to the workflow and models outlined in the paper, more clearly identify what are the “inputs” (existing data and published methods), what are the novel analysis components, and what are the outputs. I realize

the immensity of the problem here goes beyond the expectations of publication, so I mention this only as a suggestion. I strongly suggest the authors consider a tool such as `drake`, <https://docs.ropensci.org/drake/>, which is designed precisely to make these kinds of analyses more tractable and reproducible. It may at least be of use to the authors in the future in attempting to effectively package this kind of work.

We appreciate the reviewer's suggestions regarding the structure of the supporting code and agree that it needed to be more logically organized. As such, we reorganized the code from start to finish and publicly shared the revised data inputs and outputs (from the analysis) on figshare.com and code via github ([https://github.com/kholsman/EBM\\_Holsman\\_NatComm](https://github.com/kholsman/EBM_Holsman_NatComm)) We hope these changes increase transparency and facilitate downloading and running the code to regenerate analyses and figures for the paper with more ease. We feel these changes have greatly improved our code and paper and we hope will address the reviewer's suggestion. These changes include:

- 1) Reorganization of the project folder structure, following the drake template, although we don't specifically use the last "plan()" step of the drake package as that package would require users to be familiar with drake. We did however like the structure and approach and thus have organized the code into the following R files:
  - a) "R/packages.R": loads (or installs if not yet installed) all required packages,
  - b) "R/setup.R": load switches and controls,
  - c) "R/load\_functions.R": loads all the functions, now available as individual R scripts in the "R/sub\_fun" folder and creates individual figure functions (e.g., "fig2()"),
  - d) R/load\_data.R: loads all intermediate Rdata files from the "data/in" and and final Rdata files from the "data/out" folder,
  - e) "R/make.R" : runs scripts a-d.
- 2) Data was organized into two folders:
  - a) "data/in":
    - i) ACLIM simulation output from the multispecies model: "multispp\_cap\_simulations.Rdata" and "multispp\_nocap\_simulations.Rdata" as well as "0\_5\_3\_nohcr\_simulations.Rdata" generated for the broader ACLIM multimodel simulation (Hollowed et al. 2020).
    - ii) "covariates.Rdata" which include the ROMSNPZ model based annual ACLIM indices used for the hindcast and projections. These indices can also be explored interactively at: <https://kholsman.shinyapps.io/aclim/>
    - iii) target\_B\_0.Rdata and target\_B\_2.Rdata, which are the B0 and B40 targets used to generate ABC, included here for information and supplementary figures.
    - iv) Run\_Defintions.xls which include definitions of the "switches" in each of the simulations (note this also includes simulation definitions from the broader ACLIM simulation set but not presented specifically in this paper (e.g., single species simulations).
  - b) "data/out":
    - i) "Data\_defs.xls" brief definitions of each object in the Rdata files included in the out folder.
    - ii) "Biomass\_thresholds.Rdata", which are the thresholds and tipping points in fished and unfished biomass that results from the threshold analysis.
    - iii) "Catch\_thresholds.Rdata", which are the thresholds and tipping points in catch/harvest that results from the threshold analysis.

- iv) "risk.data" , which are the results of the risk analyses for a change in biomass and catch of -10%, -50%, or -80% associated with future climate conditions with or without the 2MT cap.
- 3) Added "R/sub\_scripts/make\_plots.R" which will re-generate and re-save figures 2-S6 from the paper.
- 4) Added "R/sub\_scripts/SUB\_EBM\_paper.R" which will re-generate the analyses of the paper including risk and threshold/tipping point analyses.
- 5) Updated the github README / "README\_Holsman\_EBMpaper.pdf"/.rmd" to:
  - a) include instructions to download the github repo and data (from figshare) through R,
  - b) load individual figures using functions like "fig2()" with or without overwriting existing figures in the "Figures/ folder",
  - c) Regenerate the core analyses of the paper.

We hope these changes significantly improve the ability of users to see the structure of the supporting scripts and data and welcome any additional comments or suggestions for further improving the code as needed.

Reviewers' comments, second round

Reviewer #1 (Remarks to the Author):

This is a revision of a paper looking at the performance of existing EBFM rules for Bering Sea under predicted climate trajectories. The methods used (such as bias correction etc) is appropriate and the work represents a very useful step forward for the field.

This revision has largely addressed the changes requested in the first review and is suitable for publication. A few option/minor comments are included below.

Line 6, pg 4: Perhaps put in a \$ sign to say "...and \$1.34 billion USD..."

Lines 4-5, pg 11 "...cap stabilized fisheries in the near-term but increased the risk of spawning stock biomass collapse of pollock at the end of the century...": It would be good (if possible) to explain why that is. Simply saying the threshold is breached doesn't really explain to me why it increases risk vs no cap in those circumstances. I am likely missing something obvious, but even if that is the case it might be good to lay it out clearly for the % of the readership who misses the reasoning in the way I have.

Lines 11-12, pg 11 "... 2 MT cap on harvest, also impart benefits to future fisheries through stabilizing catch...": Given the issues you identified in the previous paragraphs, perhaps tweak this to read "... 2 MT cap on harvest, also impart benefits to future fisheries through stabilizing catch, at least to a point. It appears new methods, or potentially adaptive methods may be required as systems are put under increasing stress beyond mid-century."

Line 5-10, pg 16 "... This is because the estimates of recruitment and spawning biomass from CEATTLE for the BSAI are very precise.": I note the author's offer to change terminology away from MSE. I am not asking for that, but I will also say I am not sold in the justification given here. The point of MSE is not just about precision, but also accuracy. The MSE needs to reflect as much as possible uncertainties in both properties (while I don't think this is the case here, you can be "very precisely inaccurate", as one of the authors of this paper has stressed in publications previously). I fully acknowledge that it can be hard to do this comprehensively in multispecies and ecosystem models, and so in this instance I am not going to be overly obstinate about the terminology used given the actual simulation decision, but would appreciate a line of caution about the implications of what was done vs "full MSE". For example you state, "...the effect would have been minor compared to the investigated uncertainties...". Is this because ran some preliminary tests or was it inferred due to the argument regarding precision? [ Which I note is against another model, not data.]

Line 4-5, pg 18: A minor typo (duplicate year), I believe it should read "... to model "realized catch" in each simulation year as a function..."

Line 12-15, pg 19 "Ftarget was then obtained by projecting the model forward to iteratively find the climate-informed harvest rate that corresponds to 40% of B0 (i.e., B40) with the constraint that spawning biomass during the projection period remain above 35% of B0": I assume you never hit a situation where that wasn't possible?

Line 13-14, pg 22 "... while the remaining non-CEATTLE species  $[(ABC)]_{(j,y)}$  were set to the historical (1992-2017) average for each species in the set": I still think it would be good to have a statement on how feasible this assumption is, not saying you should change it, just another note so the reader is informed on potential hiccups. I thought perhaps you could look at trajectories from EwE or FEAST as your check, do many of them stay stable or are they just as likely to change away from current/baseline values under climate? A written note here is sufficient, there is no need for plots.

pg 34: I think it should be Figures S1-S8 (not S5). Also the ordering of the supp figures is a little odd vs where they are referred to in the main text – assuming my find function is working I think

they are current referred to in the order: S7, S2, S3, S4, S5, S6, S1, S8.

Fig S2: If possible it would be nice to have the  $F=0$  case plotted on for reference. Also I like what you're trying to do here, but to help differentiate cases, does it make it illegible if "no cap2" is done with dashed lines? Also explain what no cap 2 is in the caption.

Fig S4: perhaps dash no cap cases to help differentiate them? Or pick slightly different colour set for them? Unfortunately, hue blind people struggle with the current set.

Beth Fulton

Reviewer #2 (Remarks to the Author):

The authors have thoughtfully addressed each of my concerns. I think this is an impressive body of work and appreciate their attention to detail. I believe this will be an important and well-cited contribution to Nature Communications.

July 30, 2020  
Reviewer responses to  
“Ecosystem-based fisheries management forestalls climate-driven collapse”  
K. Holsman et al.

**Reviewer comments in bold;** responses not-bolded

---

**Reviewer #1 (Remarks to the Author):**

**This is a revision of a paper looking at the performance of existing EBFM rules for Bering Sea under predicted climate trajectories. The methods used (such as bias correction etc) is appropriate and the work represents a very useful step forward for the field.**

**This revision has largely addressed the changes requested in the first review and is suitable for publication. A few option/minor comments are included below.**

**Line 6, pg 4: Perhaps put in a \$ sign to say “...and \$1.34 billion USD...”**  
Suggested change made.

**Lines 4-5, pg 11 “...cap stabilized fisheries in the near-term but increased the risk of spawning stock biomass collapse of pollock at the end of the century...”: It would be good (if possible) to explain why that is. Simply saying the threshold is breached doesn’t really explain to me why it increases risk vs no cap in those circumstances. I am likely missing something obvious, but even if that is the case it might be good to lay it out clearly for the % of the readership who misses the reasoning in the way I have.**

We appreciate this comment and agree and have tried to clarify and elaborate (briefly) this section with the following text:

*“...Indeed, we found that the EMFB 2 MT cap stabilized fisheries in the near-term but increased the risk of sudden collapse of pollock catch at the end of the century (i.e., collapse occurs rapidly without preceding declines in catch). Concurrent climate-driven declines in spawning stock biomass preceded collapse, reinforcing the importance of fishery-independent estimates of biomass for early warnings of impending fishery declines. Hyperstability is an inherent risk in decoupled harvest and biomass dynamics (Hilborn and Walters 1992) and is a potential outcome of the EBFM cap which should be explored in future MSEs...”*

We hope these changes address the reviewer’s comment.

Lines 11-12, pg 11 "... 2 MT cap on harvest, also impart benefits to future fisheries through stabilizing catch...": Given the issues you identified in the previous paragraphs, perhaps tweak this to read "... 2 MT cap on harvest, also impart benefits to future fisheries through stabilizing catch, at least to a point. It appears new methods, or potentially adaptive methods may be required as systems are put under increasing stress beyond mid-century."

Suggested change was made to the text.

Line 5-10, pg 16 "... This is because the estimates of recruitment and spawning biomass from CEATTLE for the BSAI are very precise.": I note the author's offer to change terminology away from MSE. I am not asking for that, but I will also say I am not sold in the justification given here. The point of MSE is not just about precision, but also accuracy. The MSE needs to reflect as much as possible uncertainties in both properties (while I don't think this is the case here, you can be "very precisely inaccurate", as one of the authors of this paper has stressed in publications previously). I fully acknowledge that it can be hard to do this comprehensively in multispecies and ecosystem models, and so in this instance I am not going to be overly obstinate about the terminology used given the actual simulation decision, but would appreciate a line of caution about the implications of what was done vs "full MSE". For example you state, "...the effect would have been minor compared to the investigated uncertainties...". Is this because ran some preliminary tests or was it inferred due to the argument regarding precision? [ Which I note is against another model, not data.]

We appreciate the reviewer's comment and agree with the points made here. We have updated the text slightly (additions/changes underlined) to reflect this comment and highlight future directions using full MSE approaches:

*"The MSE does not account for estimation error (uncertainty in the parameters of the operating model) nor error resulting from observation error (which will lead to the quantities on which management decisions are based differing from the operating model values). This is because the estimates of recruitment and spawning biomass from CEATTLE for the BSAI are very precise (see Figure 10 in 63) and the estimation and operating models are very similar. Thus, CEATTLE is the operating model for this MSE and implicitly the estimation method. In this approach we assume that while allowing for observation error would have increased overall error, the effect would have been minor compared to the investigated uncertainties. Future analyses using a full MSE (i.e., separate operating and estimation models) could evaluate the effect of observation error, but perhaps more importantly, the potential for model error, whereby the population dynamics model on which the estimation method is based differs from that of the operating model such the estimates on which management decisions are made are biased relative to the true values in the operating model."*

We hope these changes highlight to readers that 1) we did not evaluate the effect of model or observation error on the outcomes of this simulation and 2) future directions should evaluate our assumptions, namely that the potential for observation and model misspecification error to bias the results of this analyses was assumed low relative to climate and management effects we evaluated. We also hope this addresses the reviewer's concern regarding this topic and provides a clear path forward for future analyses.

**Line 4-5, pg 18: A minor typo (duplicate year), I believe it should read "... to model "realized catch" in each simulation year as a function..."**

Thank you for this comment and the duplicate "year" was removed.

**Line 12-15, pg 19 "Ftarget was then obtained by projecting the model forward to iteratively find the climate-informed harvest rate that corresponds to 40% of B0 (i.e., B40) with the constraint that spawning biomass during the projection period remain above 35% of B0": I assume you never hit a situation where that wasn't possible?**

We appreciate the reviewer's attention to detail here and apologize as the text was slightly incorrect as written (reflecting an older version of the text). We have corrected the text and clarified the section.

In the intermediate Ftarget determination step (but not the final Fabc step which uses the sloping HCR),  $B/B_0 < 35\%$  in the first years of the simulation for pollock because mean historical  $F < F_{40\%}$  and the increase in  $F$  in the simulation first results in a sharp increase in harvest and then a decrease in spawning biomass before coming to equilibrium (this can be seen in first few years of the corresponding figure, now SFig 7; note that this is not observed in the 2 MT cap simulations because the cap effective  $F$  rate matches historical  $F$  more closely). The sloping harvest control rule is applied (in both cap and no cap simulations) after this step to reduce  $F_{abc}$  from  $F_{40\%}$  when  $B_y < 40\%B$ , often caused in the later simulation years by climate-driven declines in both cap and no cap simulations. In addition to clarifying the text we updated the figure to illustrate this more clearly.

We would also like to note that missing from this section previously was a nuance;  $F_{40\%}$  is determined simultaneously for pollock and pcod and then for ATF (to minimize predation release, again a Council approved precautionary approach (Holsman et al. 2019) which is discussed in more detail in Holsman et al. 2016 (figure 10)). We have elaborated and updated the text in this section to clarify this (and to avoid the need to go to Holsman et al. 2016 and/or the Holsman et al. 2019 assessment for details). Thank you for renewing our attention to this section of the methods and we hope our edits are acceptable and clarify our approach.

**Line 13-14, pg 22 "... while the remaining non-CEATTLE species  $[ABC]_{(j,y)}$  were set to the historical (1992-2017) average for each species in the set": I still**

**think it would be good to have a statement on how feasible this assumption is, not saying you should change it, just another note so the reader is informed on potential hiccups. I thought perhaps you could look at trajectories from EwE or FEAST as your check, do many of them stay stable or are they just as likely to change away from current/baseline values under climate? A written note here is sufficient, there is no need for plots.**

More complex ecosystem models do show divergence in trajectories from historical averages due but those models differ significantly from CEATTLE in their method of climate-coupling and internal dynamics. Thus, although useful for placing our results in a broader context, they are not easily used to inform non-CEATTLE species ABCs. To acknowledge this and future directions we added the following text to the methods:

*“Under this approach, TAC and catch estimates from ATTACH change solely in response to changes in pollock, Pacific cod, and arrowtooth flounder input ABCs. This assumption is significant but difficult to assess without considering alternative models that resolve more species managed under the 2 MT cap along with their biological interactions and potential future changes in their relative values. That said, the three species in CEATTLE are known to strongly interact, account for a large proportion of total EBS fish biomass, and strongly drive TAC allocation. Future sensitivity analyses, possibly using more speciose food web models32,69 would be useful to quantify this sensitivity.”*

32 Reum et al. 2020. Ensemble Projections of Future Climate Change Impacts on the Eastern Bering Sea Food Web Using a Multispecies Size Spectrum Model. *Frontiers in Marine Science* 7 doi: fmars.2020.00124

69 Whitehouse and Aydin (2020) Assessing the sensitivity of three Alaska marine food webs to perturbations: an example of Ecosim simulations using Rpath. *Ecological Modelling*, 2020, vol. 429, issue

**pg 34: I think it should be Figures S1-S8 (not S5). Also the ordering of the supp figures is a little odd vs where they are referred to in the main text – assuming my find function is working I think they are current referred to in the order: S7, S2, S3, S4, S5, S6, S1, S8.**

That is correct, we updated the Supplemental figures list and updated the order of the supplemental figures.

**Fig S2: If possible it would be nice to have the F=0 case plotted on for reference. Also I like what you’re trying to do here, but to help differentiate cases, does it make it illegible if “no cap2” is done with dashed lines? Also explain what no cap 2 is in the caption.**

Thank you for finding the type-o in the legend, that has been corrected to read “No cap” and “2 MT cap” accordingly. We have also corrected the scales which should have been

divided by  $1e6$ . We thank the reviewer for the suggestion of additional approaches to displaying Fig S2. We evaluated dashed lines but ultimately dismissed this approach as it was our opinion that the pattern was too hard to distinguish with dashes due to annual variability in the patterns. Similarly, although we attempted a few different approaches, we ultimately returned to this version as we found that overlaying  $F=0$  on Fig S2 obfuscates the patterns and key message of the figure. We note that  $F = 0$  is plotted in Figure 3 facilitating a side by side comparison. It is our feeling that this is the clearest way to display the multiple scenarios and support the corresponding points in the text but we will defer to the editorial staff on this matter.

**Fig S4: perhaps dash no cap cases to help differentiate them? Or pick slightly different colour set for them? Unfortunately, hue blind people struggle with the current set.**

We appreciate the comment regarding this figure and have increased the contrast in coloration to help distinguish near-term from long-term. Please note that “2 MT cap” and “no cap” are distinguished as circle and square already, dashed lines do not apply to this figure.

**Beth Fulton**

**Reviewer #2 (Remarks to the Author):**

**The authors have thoughtfully addressed each of my concerns. I think this is an impressive body of work and appreciate their attention to detail. I believe this will be an important and well-cited contribution to Nature Communications.**

Thank you and we are pleased to hear that our changes sufficiently addressed the reviewer’s comments.